# Neural optimal feedback control
# with local learning rules

**Johannes Friedrich** [1] **Siavash Golkar** [1] **Shiva Farashahi** [1]

**Alexander Genkin** [5] **Anirvan M. Sengupta** [2,3,4] **Dmitri B. Chklovskii** [1,5]

[1] Center for Computational Neuroscience, Flatiron Institute
[2] Center for Computational Mathematics, Flatiron Institute
[3] Center for Computational Quantum Physics, Flatiron Institute
[4] Department of Physics and Astronomy, Rutgers University
[5] Neuroscience Institute, NYU Medical Center

`{jfriedrich,sgolkar,sfarashahi,dchklovskii}@flatironinstitute.org`
`{alexander.genkin,anirvans.physics}@gmail.com`

## Abstract

A major problem in motor control is understanding how the brain plans and executes proper movements in the face of delayed and noisy stimuli. A prominent framework for addressing such control problems is Optimal Feedback Control (OFC). OFC generates control actions that optimize behaviorally relevant criteria by integrating noisy sensory stimuli and the predictions of an internal model using the Kalman filter or its extensions. However, a satisfactory neural model of Kalman filtering and control is lacking because existing proposals have the following limitations: not considering the delay of sensory feedback, training in alternating phases, and requiring knowledge of the noise covariance matrices, as well as that of systems dynamics. Moreover, the majority of these studies considered Kalman filtering in isolation, and not jointly with control. To address these shortcomings, we introduce a novel online algorithm which combines adaptive Kalman filtering with a model free control approach (i.e., policy gradient algorithm). We implement this algorithm in a biologically plausible neural network with local synaptic plasticity rules. This network performs system identification and Kalman filtering, without the need for multiple phases with distinct update rules or the knowledge of the noise covariances. It can perform state estimation with delayed sensory feedback, with the help of an internal model. It learns the control policy without requiring any knowledge of the dynamics, thus avoiding the need for weight transport. In this way, our implementation of OFC solves the credit assignment problem needed to produce the appropriate sensory-motor control in the presence of stimulus delay.

## 1 Introduction

The sensorimotor control system has exceptional abilities to perform fast and accurate movements in a variety of situations. To achieve such skillful control, this system faces two key challenges: (i) sensory stimuli are noisy, making estimation of current state of the system difficult, and (ii) sensory stimuli is often delayed, which if unaccounted, results in movements that are inaccurate and unstable [1]. Optimal Feedback Control (OFC) has been proposed as a solution to this control problem [2, 3]. OFC approaches these problems by building an internal model of the system dynamics, and using

35th Conference on Neural Information Processing Systems (NeurIPS 2021).

this internal model to generate control actions. OFC often employs Kalman filtering to optimally integrate the predictions of this internal model and the noisy/delayed sensory stimuli.

Because of the power and flexibility of the OFC framework, biologically plausible neural architectures capable of building such internal models has been under active investigation. Specifically, earlier works used attractor dynamics implemented through a recurrent basis function network [4] or a line attractor network [5] to implement Kalman filters. Kalman filtering and control has also been implemented through different phases of estimation, system identification and control [6], and more recently, using a particle filtering method for Kalman filtering [7].

Nonetheless, these works suffer from major limitations. Importantly, none of these considered that sensory feedback is delayed [4, 6, 5, 7, 8], although it has been prominent in the original computational-level OFC proposal [2], or merely considered the case of Kalman filtering, and not the combination of it with control [4, 5, 7, 8]. These works also required knowledge of the noise covariances, either a priori [4, 5, 7, 8] or obtained in a separate 'offline sensor' mode [6]. Moreover, many of these works lack biological plausibility and realism one would expect from a viable model of brain function [4, 5, 7, 8]. Crucially, biological plausibility requires the network to operate online, (i.e. receive a stream of noisy measurement data and process them on the fly), and also requires synaptic plasticity rules to be local (i.e. learn using rules that only depend on variables represented in pre and postsynaptic neurons and/or on global neuromodulatory signals). Lastly, several of these models suffer from combinatorial explosion as the dimensionality of the input grows [4, 5], require running an inner loop until convergence at each time step [8, 6], or require separate learning and execution phases [6], cf. Table 1.

We address these shortcomings and present a complete neural implementation of optimal feedback control, thus tackling an open issue in biological control [9]. In this model, which we call Bio-OFC, the state space, the prediction error [10, 11] (i.e., the mismatch between the network's internal prediction and delayed sensory feedback), and the control are represented by different neurons, cf. Fig. 1. The network also receives scalar feedback related to the objective function, as a global signal, and utilizes this signal to update the synaptic connection according to policy gradient method [12, 13]. To test the performance of our network, we simulate Bio-OFC in episodic (finite horizon) tasks (e.g., a discrete-time double integrator model, a hand reaching task [1], and a simplified fly simulation).

**Summary of contributions:**

- We introduce Bio-OFC, a biologically plausible neural network that combines adaptive model based state discovery via adaptive Kalman filtering with a model free control agent.

- Our implementation does not require knowledge of noise covariances nor the system dynamic, considers delayed sensory feedback, and has no separate learning/execution phases.

- Our model-free control agent enables closed-loop control, thus avoiding the weight transport problem, a challenging problem even in non-biological control. [14, 15]

Table 1: **Limitations of previously proposed neural implementations of OFC.** Presence or absence of different properties in previously proposed neural models, and their comparison to Bio-OFC. Guide to symbols: ✓: true, ✗: false, ✓✗: partially true, N/A: not applicable.

| | [4] | [6] | [5] | [7] | [8] | Bio-OFC |
|---|---|---|---|---|---|---|
| delayed sensory feedback | ✗ | ✗ | ✗ | ✗ | ✗ | ✓ |
| control included | ✗ | ✓ | ✗ | ✗ | ✗ | ✓ |
| noise covariance agnostic | ✗ | ✓ | ✗ | ✗ | ✗ | ✓ |
| online system identification | ✗ | ✓ | ✗ | ✓ | ✓✗ | ✓ |
| local learning rules | N/A | ✓ | N/A | ✗ | ✓ | ✓ |
| tractable latent size | ✗ | ✓ | ✗ | ✓ | ✓ | ✓ |
| absence of inner loop | ✓ | ✗ | ✓ | ✓ | ✗ | ✓ |
| single phase learning/execution | N/A | ✗ | N/A | ✓ | ✓ | ✓ |

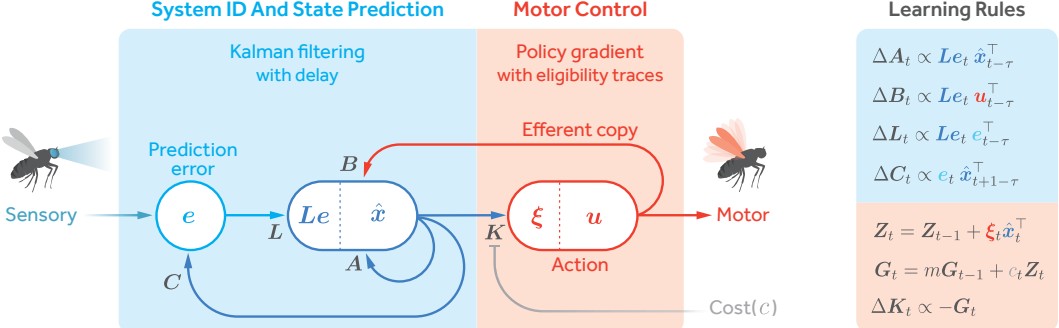

Figure 1: **The circuit and learning rules of the Bio-OFC algorithm.** Our circuit is comprised of two main parts. First (in blue), the circuit performs Kalman filtering. Then (in red), the circuit performs control using policy gradients with eligibility traces. Triangular arrowheads denote synaptic connections and the flat arrowhead denotes the modulatory effect of the cost signal.

## 2 Background

We review classical Kalman estimation and control in this section, using boldface lowercase/uppercase letters for vectors/matrices and $\boldsymbol{I}$ for the identity matrix.

### 2.1 Problem formulation

We model the environment as a linear dynamical system driven by control input and perturbed by Gaussian noise. The true state of the system $\boldsymbol{x}$ is hidden and all the animal has access to are the observations $\boldsymbol{y}$ that are assumed to be linear functions of the state corrupted by Gaussian noise.

$$\text{dynamics:} \qquad \boldsymbol{x}_{t+1} = \boldsymbol{A}\boldsymbol{x}_t + \boldsymbol{B}\boldsymbol{u}_t + \boldsymbol{v}_t \tag{1}$$
$$\text{observation:} \qquad \boldsymbol{y}_t = \boldsymbol{C}\boldsymbol{x}_t + \boldsymbol{w}_t \tag{2}$$

Here $\boldsymbol{v}_t \sim \mathcal{N}(0; \boldsymbol{V})$ and $\boldsymbol{w}_t \sim \mathcal{N}(0, \boldsymbol{W})$ are independent Gaussian random variables and the initial state has a Gaussian prior distribution $\boldsymbol{x}_0 \sim \mathcal{N}(\hat{\boldsymbol{x}}_0, \boldsymbol{\Sigma}_0)$.

The goal is to estimate the latent state $\hat{\boldsymbol{x}}$ in order to design a control $\boldsymbol{u}$ that minimizes expected cost

$$\text{expected cost:} \qquad J = \mathbb{E}\left[\sum_{t=0}^{T} c(\boldsymbol{x}_t, \boldsymbol{u}_t)\right] \tag{3}$$
$$\text{control:} \qquad \boldsymbol{u}_t = k(\hat{\boldsymbol{x}}_t) = \arg\min J \tag{4}$$

where $c(\boldsymbol{x}_t, \boldsymbol{u}_t)$ is the instantaneous cost associated with state $\boldsymbol{x}_t$ and action $\boldsymbol{u}_t$. As the environment dynamics is not known to the animal a priori, the parameters $\boldsymbol{A}, \boldsymbol{B}, \boldsymbol{C}$ must be learned online.

### 2.2 Kalman estimation and control

The Kalman filter [16, 2] is an estimator of the latent state $\hat{\boldsymbol{x}}_t$ (and its variance) via a weighted summation of the current observation and the prediction of the internal model based on prior measurements. However, in biologically realistic situations the sensory feedback is always delayed. This means that the control signal $\boldsymbol{u}_t$ has to be issued, and thus the state $\boldsymbol{x}_t$ estimated, before $\boldsymbol{y}_t$ has been observed. The appropriately modified Kalman filter computes the posterior probability distribution of $\boldsymbol{x}_t$ given observations $\boldsymbol{y}_{t-\tau}, ..., \boldsymbol{y}_0$ where $\tau \geq 1$ is the delay. We start here with the case of $\tau = 1$ for which the recursive updates of the mean $\hat{\boldsymbol{x}}_t$ and the variance $\boldsymbol{\Sigma}_t$ are

$$\hat{\boldsymbol{x}}_{t+1} = \boldsymbol{A}\hat{\boldsymbol{x}}_t + \boldsymbol{B}\boldsymbol{u}_t + \boldsymbol{L}_t(\boldsymbol{y}_t - \boldsymbol{C}\hat{\boldsymbol{x}}_t) \tag{5}$$
$$\boldsymbol{\Sigma}_{t+1} = (\boldsymbol{A} - \boldsymbol{L}_t\boldsymbol{C})\boldsymbol{\Sigma}_t\boldsymbol{A}^\top + \boldsymbol{V} \tag{6}$$

where $\boldsymbol{L}_t$ is known as the Kalman gain matrix which optimally combines the noisy observations $\boldsymbol{y}_t$ with the internal model and is given by

$$\boldsymbol{L}_t = \boldsymbol{A}\boldsymbol{\Sigma}_t\boldsymbol{C}^\top(\boldsymbol{C}\boldsymbol{\Sigma}_t\boldsymbol{C}^\top + \boldsymbol{W})^{-1}. \tag{7}$$

The Kalman filter is optimal in the sense that it minimizes the mean-squared error $\mathbb{E}[e_t \top e_t]$ with prediction error (innovation) $e_t = y_t - C\hat{x}_t$.

The output feedback law, also known as policy, (4) simplifies if the cost $J$ is quadratic in $u_t$ and $x_t$:

$$u_t = -K\hat{x}_t \tag{8}$$

and is known as linear-quadratic regulator (LQR). The control gain, $K$, is found by solving a matrix Riccati equation (cf. Supplementary Material). Linear policies have been successfully applied to a variety of control tasks [17].

## 3 Neural network representation for optimal feedback control

### 3.1 Inference

For now, let us assume that the system dynamics, the Kalman gain and the control gain $A, B, -C$, $L$, and $K$ are constant and known. Then, the latent state $\hat{x}$ can be obtained by the Kalman estimator Eq. (5). This algorithm naturally maps onto the network in Fig. 1 and Supplementary Fig. S1A with neural populations representing $\hat{x}$, $e := y - C\hat{x}$, and $u$ that are connected by synapses whose weights represent the elements of the matrices $A, B, -C$ and $L$. The computation of the control variable $u$ according to Eq. (8) can be implemented by synapses whose weights represent the elements of the matrix $-K$.

If the sensory stimulus delay is $\tau = 1$, our network implements the Kalman prediction reviewed in the previous section. In case $\tau > 1$, the latent state must be recomputed throughout the delay period which requires a biologically implausible circuit (cf. Supplementary Fig. S1C). To overcome this, we adapt an alternative solution from online control [18, 19]. Specifically, we combine delayed measurements with the similarly delayed latent state estimation. Such inference can be performed by the network of the same architecture and adjusting the synaptic delay associated with matrix $C$ to match with the sensory delay, in accordance with the following expression:

$$\hat{x}_{t+1} = A\hat{x}_t + Bu_t + L\underbrace{(y_{t+1-\tau} - C\hat{x}_{t+1-\tau})}_{e_t} \tag{9}$$

As we show in Results (Fig. 3) this reduces predictive performance only modestly compared to the biologically unrealistic scheme. More details on the inference process and the temporal order in which our recurrent network performs the above steps is shown in Supplementary Fig. S1B.

### 3.2 Learning

#### 3.2.1 System identification and Kalman gain

Next, we turn our attention to the system identification/learning problem, which was not addressed by some of the previous proposals, cf. Table 1. Given a sequence of observations $\{y_0, \cdots, y_T\}$, we use a least squares approach [20] to find the parameters that minimize the mean-square prediction error $\frac{1}{T} \sum_{t=0}^{T} e_t^\top e_t$. We perform this optimization in an online manner, that is at each time-step $t + 1$, after making the delayed observation $y_{t+1-\tau}$, we update the parameter estimates $\hat{A}, \hat{B}, \hat{C}$ using steps that minimize $e_t^\top e_t$, assuming that the state estimate and actions corresponding to prior observations (e.g. $\hat{x}_{t-\tau}, u_{t-\tau}$ etc.) are fixed. To obtain $L$ we would like to avoid solving the Riccati equation (6) as it requires matrix operations difficult to implement in biology. So, we use the same optimization procedure to update the Kalman gain $L$. Using Eq. (9) and explicitly writing out the matrix/vector indices as superscripts yields the following stochastic gradient with respect to $A$:

$$-\frac{\partial}{\partial A^{ij}} \frac{1}{2} \sum_k \left(e_t^k\right)^2 = -\sum_k e_t^k \frac{\partial e_t^k}{\partial A^{ij}} = -\sum_k e_t^k \frac{\partial \left(y_{t+1-\tau}^k - \sum_l C^{kl}\hat{x}_{t+1-\tau}^l\right)}{\partial A^{ij}} =$$

$$= \sum_{k,l} e_t^k C^{kl} \frac{\partial \left(\sum_m A^{lm}\hat{x}_{t-\tau}^m + \sum_n B^{ln}u_t^n + \sum_p L^{lp}e_t^p\right)}{\partial A^{ij}} =$$

$$= \sum_{k,l,m} e_t^k C^{kl}\delta^{li}\delta^{mj}\hat{x}_{t-\tau}^m = \sum_k e_t^k C^{ki}\hat{x}_{t-\tau}^j \tag{10}$$

Performing similar derivations for the other synaptic weights, our optimization procedure would rely on the following stochastic gradients:

$$-\nabla_{\boldsymbol{A}} \tfrac{1}{2} \boldsymbol{e}_t^\top \boldsymbol{e}_t = \boldsymbol{C}^\top \boldsymbol{e}_t \hat{\boldsymbol{x}}_{t-\tau}^\top \qquad\qquad -\nabla_{\boldsymbol{B}} \tfrac{1}{2} \boldsymbol{e}_t^\top \boldsymbol{e}_t = \boldsymbol{C}^\top \boldsymbol{e}_t \boldsymbol{u}_{t-\tau}^\top \qquad (11)$$

$$-\nabla_{\boldsymbol{L}} \tfrac{1}{2} \boldsymbol{e}_t^\top \boldsymbol{e}_t = \boldsymbol{C}^\top \boldsymbol{e}_t \boldsymbol{e}_{t-\tau}^\top \qquad\qquad -\nabla_{\boldsymbol{C}} \tfrac{1}{2} \boldsymbol{e}_t^\top \boldsymbol{e}_t = \boldsymbol{e}_t \hat{\boldsymbol{x}}_{t+1-\tau}^\top \qquad (12)$$

This yields a classical Hebbian rule between (a memory trace of) presynaptic activity $\hat{\boldsymbol{x}}_{t+1-\tau}$ and postsynaptic activity $\boldsymbol{e}_t$ for weights $\boldsymbol{C}$. However, it suggests non-local learning rules for $\boldsymbol{A}, \boldsymbol{B}, \boldsymbol{L}$, which runs contrary to biological requirements. We can circumvent this problem by replacing $\boldsymbol{C}^\top$ with $\boldsymbol{L}$, which corresponds to left-multiplication of the gradients with a positive definite matrix (see Supplementary Material Sec. 4). This still decreases the mean-square prediction error under some mild initialization constraints on $\boldsymbol{C}$ and $\boldsymbol{L}$ and yields local plasticity rules, cf. Fig. 1,

$$\Delta \hat{\boldsymbol{A}}_t \propto \boldsymbol{L} \boldsymbol{e}_t \, \hat{\boldsymbol{x}}_{t-\tau}^\top \qquad (13) \qquad\qquad \Delta \hat{\boldsymbol{B}}_t \propto \boldsymbol{L} \boldsymbol{e}_t \, \boldsymbol{u}_{t-\tau}^\top \qquad (15)$$

$$\Delta \boldsymbol{L}_t \propto \boldsymbol{L} \boldsymbol{e}_t \, \boldsymbol{e}_{t-\tau}^\top \qquad (14) \qquad\qquad \Delta \hat{\boldsymbol{C}}_t \propto \boldsymbol{e}_t \, \hat{\boldsymbol{x}}_{t+1-\tau}^\top \qquad (16)$$

where the input current $\boldsymbol{L}\boldsymbol{e}_t$ is locally available at neurons representing $\hat{\boldsymbol{x}}$. The first three rules are local, but non-Hebbian, capturing correlations between presynaptic activity $\hat{\boldsymbol{x}}_{t-\tau}, \boldsymbol{u}_{t-\tau}, \boldsymbol{e}_{t-\tau}$ and postsynaptic current $\boldsymbol{L}\boldsymbol{e}_t$. Note that these updates do not require knowledge of the noise covariances $\boldsymbol{V}$ and $\boldsymbol{W}$, an advantage over previous work, cf. Table 1.

### 3.2.2 Control

Next, we consider optimal control a neural implementation of which was missing in most previous proposals, cf. Table 1. Traditionally, optimal control law is computed by iterating a matrix Riccati equation, cf. Supplementary Material, posing a difficult challenge for a biological neural implementation (but see [21]). To circumvent this problem, we propose to learn the controller weights $\boldsymbol{K}$ using a policy gradient method [12, 22] instead. Policy gradient methods directly parametrize a stochastic controller $\pi_{\boldsymbol{K}}(\boldsymbol{u}|\boldsymbol{x})$. Representing the total cost for a given trajectory $\tau$ as $c(\tau)$, they optimize the parameters $\boldsymbol{K}$ by performing gradient descent on the expected cost $J = \mathbb{E}_{\pi_{\boldsymbol{K}}}[c(\tau)] = \mathbb{E}_{\pi_{\boldsymbol{K}}}\left[\sum_{t=0}^{T} c_t\right]$.

$$\nabla_{\boldsymbol{K}} J = \int c(\tau) \nabla_{\boldsymbol{K}} \pi_{\boldsymbol{K}}(\tau) d\tau = \mathbb{E}_{\pi_{\boldsymbol{K}}}\left[c(\tau) \nabla_{\boldsymbol{K}} \log \pi_{\boldsymbol{K}}(\tau)\right] \qquad (17)$$

$$= \mathbb{E}_{\pi_{\boldsymbol{K}}}\left[\left(\sum_{t=0}^{T} c_t\right)\left(\sum_{s=0}^{T} \nabla_{\boldsymbol{K}} \log \pi_{\boldsymbol{K}}(\boldsymbol{u}_s|\boldsymbol{x}_s)\right)\right] \qquad (18)$$

The term in square brackets is an unbiased estimator of the gradient and can be used to perform stochastic gradient decent. As already hinted at by [12], due to causality, costs $c_t$ are not affected by later controls $\boldsymbol{u}_s, s > t$, and the variance of the estimator can be reduced by excluding those terms, which yields

$$\Delta \boldsymbol{K} \propto -\sum_{t=0}^{T} c_t \left(\sum_{s=0}^{t} \nabla_{\boldsymbol{K}} \log \pi_{\boldsymbol{K}}(\boldsymbol{u}_s|\boldsymbol{x}_s)\right). \qquad (19)$$

We simply keep an eligibility trace of the past, $\boldsymbol{Z}_t = \sum_{s=0}^{t} \nabla_{\boldsymbol{K}} \log \pi_{\boldsymbol{K}}(\boldsymbol{u}_s|\boldsymbol{x}_s)$, and perform parameter updates $\Delta \boldsymbol{K} \propto -c_t \boldsymbol{Z}_t$ at each time step $t$. A similar update rule has been suggested by [23, 24] for the infinite horizon case. Global convergence of policy gradient methods for linear quadratic regulator has been recently studied by Fazel *et al.* [25].

We assume the output of neurons encoding control $\boldsymbol{u}$ is perturbed by Gaussian noise

$$\boldsymbol{u}_t = -\boldsymbol{K}\hat{\boldsymbol{x}}_t - \boldsymbol{\xi}_t \quad \text{with} \quad \boldsymbol{\xi}_t \sim \mathcal{N}(0, \sigma^2 \boldsymbol{I}). \qquad (20)$$

The synapses are endowed with a synaptic tag [26] $\boldsymbol{Z}$, an eligibility trace that tracks correlations between pre-synaptic activity $\hat{\boldsymbol{x}}$ and post-synaptic noise $\boldsymbol{\xi}$. It is reset to zero at the beginning of each trajectory, though instead of a hard reset it could also softly decay with a time constant of the same order $\mathcal{O}(T)$ as trajectory duration [27]. The weight update assigns cost $c_t$ to the synapses according to their eligibility $\boldsymbol{Z}_t$. The cost is e.g. provided by a diffuse neuromodulatory signal such as dopamine.

The optional use of momentum $m \in [0, 1)$ adds a low-pass filter to the synaptic plasticity cascade:

$$\boldsymbol{Z}_t = \boldsymbol{Z}_{t-1} + \boldsymbol{\xi}_t \hat{\boldsymbol{x}}_t^\top \quad \left( = \sigma^2 \sum_{s=0}^t \nabla_{\boldsymbol{K}} \log \pi_{\boldsymbol{K}} (\boldsymbol{u}_s | \hat{\boldsymbol{x}}_s) \right) \tag{21}$$

$$\boldsymbol{G}_t = m \boldsymbol{G}_{t-1} + c_t \boldsymbol{Z}_t \tag{22}$$

$$\Delta \boldsymbol{K}_t \propto -\boldsymbol{G}_t \tag{23}$$

## 4   Experiments

In this section we look at three different experiments to demonstrate various features of the Bio-OFC algorithm. In Sec. 4.1, we look at a discrete-time double integrator and discuss how our approach performs not only Kalman filtering (learning the optimal Kalman gain), but also full system-ID in the open-loop setting (system-ID followed by control) as well as in the more challenging closed-loop setting (simultaneous system-ID and control). In each case, we provide quantitative comparisons and discuss the effect of increased delay. In Secs. 4.2 and 4.3, we apply our methodology to two biologically relevant control tasks, that of reaching movements and flight.

### 4.1   Discrete-time double integrator

As a simple test case, we follow along the lines of [28] and consider the classic problem of a discrete-time double integrator with the dynamical model

$$\boldsymbol{x}_{t+1} \sim \mathcal{N}(\boldsymbol{A}\boldsymbol{x}_t + \boldsymbol{B}\boldsymbol{u}_t, \boldsymbol{V}) \quad \text{where} \quad \boldsymbol{A} = \begin{pmatrix} 1 & 1 \\ 0 & 1 \end{pmatrix}, \quad \boldsymbol{B} = \begin{pmatrix} 0 \\ 1 \end{pmatrix}, \quad \boldsymbol{V} = \begin{pmatrix} .01 & 0 \\ 0 & .01 \end{pmatrix}. \tag{24}$$

Such a system models the position and velocity (respectively the first and second components of the state) of a unit mass object under force $u$. As an instance of LQR, we can try to steer this system to reach point $(0, 0)^\top$ from initial condition $\boldsymbol{x}_0 = (-1, 0)^\top$ without expending much force:

$$J = \sum_{t=0}^{T} \boldsymbol{x}_t^\top \boldsymbol{Q} \boldsymbol{x}_t + R \sum_{t=0}^{T-1} u_t^2 \quad \text{where} \quad \boldsymbol{Q} = \begin{pmatrix} 1 & 0 \\ 0 & 0 \end{pmatrix}, \quad R = 1, \quad T = 10 \tag{25}$$

We assume that the initial state estimate $\hat{\boldsymbol{x}}_0$ is at the true initial state ($\hat{\boldsymbol{x}}_0 = \hat{\boldsymbol{C}}^+ \boldsymbol{C} \boldsymbol{x}_0$).

We go beyond LQR (cf. Supplementary Fig. S2A) and assume $\boldsymbol{x}$ is not directly observable (cf. Supplementary Fig. S2B,C), but we merely have access to noisy observations, $\boldsymbol{y} \sim \mathcal{N}(\boldsymbol{C}\boldsymbol{x}, \boldsymbol{W})$. We consider two observation models, one where the state is only observed with some uncorrelated noise, and one where the observation noise covariance is not diagonal and an additional mixture of the 2 state components is observed:

$$\text{LDS1:} \ \boldsymbol{C} = \begin{pmatrix} 1 & 0 \\ 0 & 1 \end{pmatrix}, \boldsymbol{W} = \begin{pmatrix} .04 & 0 \\ 0 & .25 \end{pmatrix} \qquad \text{LDS2:} \ \boldsymbol{C} = \begin{pmatrix} 1 & 0 \\ 0 & -1 \\ .5 & .5 \end{pmatrix}, \boldsymbol{W} = \begin{pmatrix} .04 & .09 & 0 \\ .09 & .25 & 0 \\ 0 & 0 & .04 \end{pmatrix} \tag{26}$$

We denote these two systems as linear dynamical systems 1 and 2 (LDS1 and LDS2).

**Learning the Kalman gain.**   A major advantage of our work is that it does not require knowledge of the covariance matrices $\boldsymbol{V}$ and $\boldsymbol{W}$ to determine the Kalman filter gain $\boldsymbol{L}$. We studied this using LDS1 in a scenario where the observation noise varies; changing the covariance matrix $\boldsymbol{W}$ from $\text{diag}(.04, .25)$ to $\text{diag}(.04, .01)$ after 2500, to $\text{diag}(.01, .01)$ after 5000, and back to $\text{diag}(.04, .25)$ after 7500 episodes. In this experiment we fix $\boldsymbol{A}, \boldsymbol{B}, \boldsymbol{C}$ at the ground truth, and initialize $\boldsymbol{L}$ at the optimal value for $\boldsymbol{W} = \text{diag}(.04, .25)$, and updated the latter according to Eq. (14). Fig. 2 shows how the elements of the filter matrix $\boldsymbol{L}$ adapt in time, to optimize performance as measured by the mean squared prediction error. The learning rate was tuned to minimize the average MSE over all episodes. Although the Kalman gain can be slow to converge in some cases (Fig. 2A), performance is quickly close to optimal (Fig. 2B).

Fig. 3 shows the achievable optimal control cost as a function of delay for four different controllers: the optimal linear-quadratic-Gaussian (LQG) controller that uses time-dependent gains, the biologically implausible ANN (cf. Supplementary Fig. S1C) that uses time-invariant gains, a model-free[1] approach

---

[1]Note that LQG uses the model for state inference as well as for control. Our Bio-OFC uses the model only to infer the latent state, but uses a model-free controller. In contrast, model-based reinforcement learning algorithms typically assume knowledge of the current state and use the model only for control.

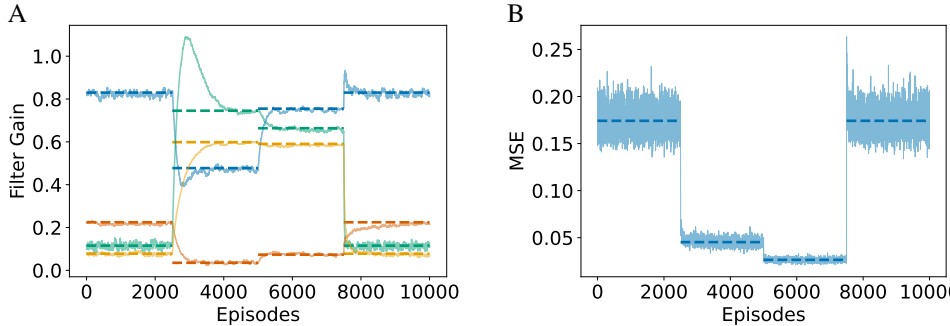

Figure 2: **Bio-OFC adapts to changing noise statistics. (A)** Filter gain (colors denote different elements of the gain matrix) and **(B)** mean squared prediction error (MSE) in the simple LQG task with 2 latent dimensions and 2-d observations (LDS1, see text for details). Solid lines show the mean over 20 runs. Dashed lines indicate the optimal filter gain and the corresponding average MSE. After 2500 episodes the observation noise covariance $W$ decreased to $\mathrm{diag}(.04, .01)$, after 5000 to $\mathrm{diag}(.01, .01)$, and after 7500 back to $\mathrm{diag}(.04, .25)$.

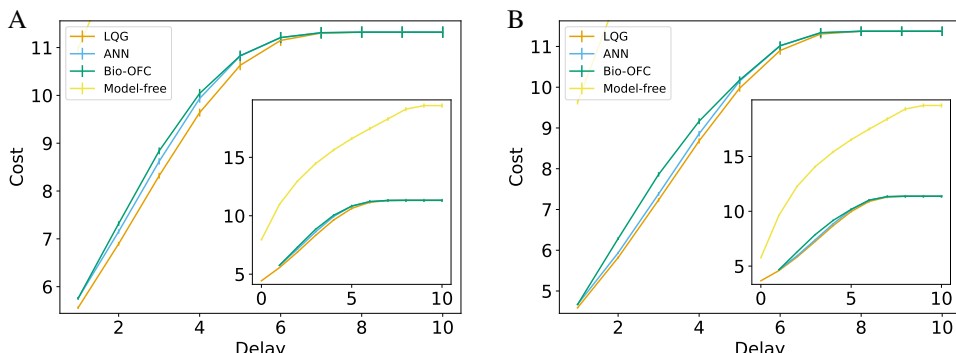

Figure 3: **Optimal cost of Bio-OFC is close to that of LQG for various delays.** Optimal cost as function of measurement delay obtained by different methods. We used the simple LQG tasks with 2 latent dimensions and **(A)** 2-d observations (LDS1, see text for details) and **(B)** 3-d observations (LDS2). LQG uses the optimal time-dependent filter and feedback gains, ANN uses the network of Supplementary Fig. S1C, Bio-OFC the network of Fig. 1 (cf. Supplementary Figs. S1B and S2D), and model-free directly maps from noisy delayed observations to control, $\boldsymbol{u}_t = \boldsymbol{K}\boldsymbol{y}_{t-\tau}$ (cf. Supplementary Fig. S2E). Shown is the mean cost $\pm$ SEM over 10000 episodes.

using policy gradient method applied directly to observations (inset), and Bio-OFC that update the current state estimate $\hat{\boldsymbol{x}}_t$ directly using the delayed measurement $\boldsymbol{y}_{t-\tau}$ (cf. Eq. (9) and Fig. 1). For the sake of biological plausibility, we imposed time-invariant gains and direct state estimate updates based on delayed measurement. These results clearly demonstrate that Bio-OFC is robust to sensory delays. Specifically, while it is expected that Bio-OFC will not learn a solution quite as good as LQG, our results show that the solution found by it is not very far off. In general, a model can be useful in two ways: It facilitates a filtered estimate that is useful even in the absence of measurement delays (see LQG vs model-free for delay 0 in Fig. 3), and in the presence of delays the model helps to bridge the gap by predicting forward in time.

**Full system identification.** We next considered the case of system identification, i.e. learning the weight matrices $\boldsymbol{A}, \boldsymbol{B}, \boldsymbol{C}$ in addition to $\boldsymbol{L}$, using Eqs. (13-16). We initialized $\boldsymbol{A}$ and $\boldsymbol{B}$ with small random numbers drawn from $\mathcal{N}(0, 0.01)$. For LDS1, $\boldsymbol{C}$ and $\boldsymbol{L}$ were initialized as diagonal dominated random matrices, with diagonal elements drawn uniformly randomly from $[0.5, 1]$ and off-diagonal ones from $[0, 0.5]$. For LDS2, they were drawn from $\mathcal{N}(0, 0.01)$ under the constraint that the symmetric part of $\boldsymbol{LC}$ has positive eigenvalues. Controls $u_t$ were drawn from $\mathcal{N}(0, 0.25)$. Fig. 4A,B show how the mean squared prediction error converges to the optimal values from Fig. 3 (dashed horizontal lines). We also considered a Bio-OFC that learns in environment LDS2, but

over-represents the latent space, assuming it is 3-d instead of 2-d. The square matrices $C$ and $L$ were initialized analogously to LDS1. Fig. 4C shows that this over-representation does not affect performance.

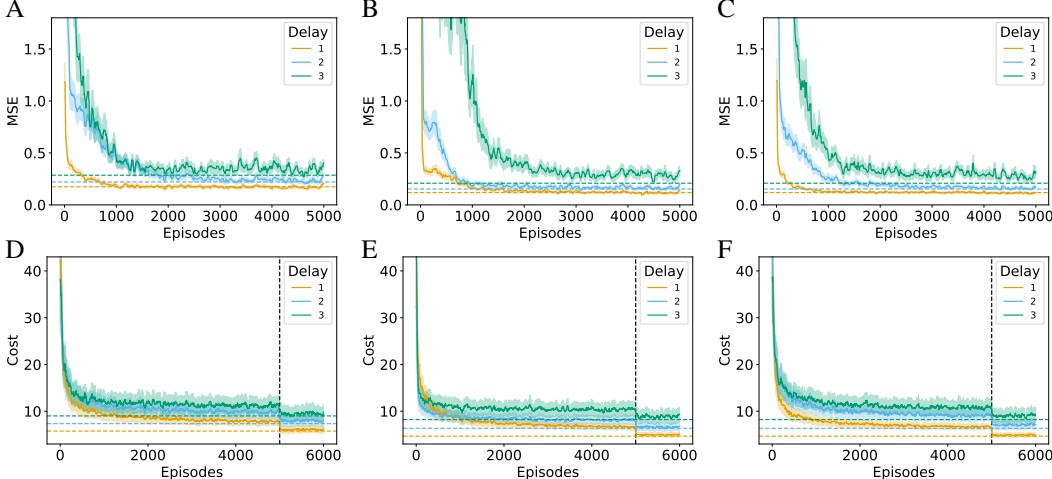

Figure 4: **Open-loop Bio-OFC converges to optimally achievable MSE and cost given the delay.** The solid (shaded) curve depicts the mean ($\pm$SEM) of 20 runs with different random initial weights, smoothed using a running median with window size 51. The dashed horizontal lines show the asymptotic values from Fig. 3. The dashed vertical line depicts the time when learning was stopped and the exploratory stochasticity in the controller removed. Mean squared prediction error during system identification, using plasticity rules Eqs. (13-16) for the filter and random Gaussian controller input, as function of episodes for **(A)** 2-d observations (LDS1), **(B)** 3-d observations (LDS2) and **(C)** 3-d observations (LDS2) with an over-representing Bio-OFC that assumes 3 instead of the actual 2 latent dimensions. Cost as function of episodes, using plasticity rules Eqs. (21-23) for the controller, for **(D)** 2-d observations (LDS1, see text for details), **(E)** 3-d observations (LDS2) and **(F)** 3-d observations (LDS2) with an over-representing Bio-OFC that assumes 3 instead of the actual 2 latent dimensions.

**System identification followed by control.** After performing system identification for 5000 episodes, we kept $A, B, C, L$ fixed and transitioned to learning the controller $K$ using Eqs. (21-23) for another 5000 episodes with controller noise $\xi \sim \mathcal{N}(0, 0.04)$, cf. Eq. (20) and Fig. 4D-F. The stochasticity in the controller results in an excess cost. Using a deterministic controller ($\xi = 0$) for another 1000 episodes reveals that the network converged to the optimal cost from Fig. 3 (dashed horizontal lines). We used two learning rates, one for $A, B, C, L$ and one for $K$ with momentum $m = 0.99$ for the latter. Learning rates that quickly yield good final performance were chosen by minimizing the sum of average reward during and after learning using Optuna [29], a hyperparameter optimization framework freely available under the MIT license. Different noise levels $\sigma$ and momenta $m$ are considered in Supplementary Figs. S6 and S7 respectively.

**Simultaneous system ID and control.** Performing system identification prior to learning a controller is known as open-loop adaptive control and a common approach in control theory. Recent advances in the control community tackle the more challenging problem of closed-loop control [14], [2] simultaneously identifying both while controlling the system. Our network is capable of closed-loop control with no separate phases for system identification and control optimization necessary. In contrast to our work, the only other proposed neural implementation that also includes control [6] requires separate phases for system-ID and control. Fig. 5 shows how the control cost evolves in time when the weights are updated according to Eqs. (13-16) and Eqs. (21-23) while using the controller designed inputs of Eq. (20). Again, using a deterministic controller ($\xi = 0$) for another 1000 episodes

---

[2]When a controller designs the inputs based on the history of inputs and observations, the inputs become highly correlated with the past process noise sequences, which prevents consistent and reliable parameter estimation with standard system identification techniques.

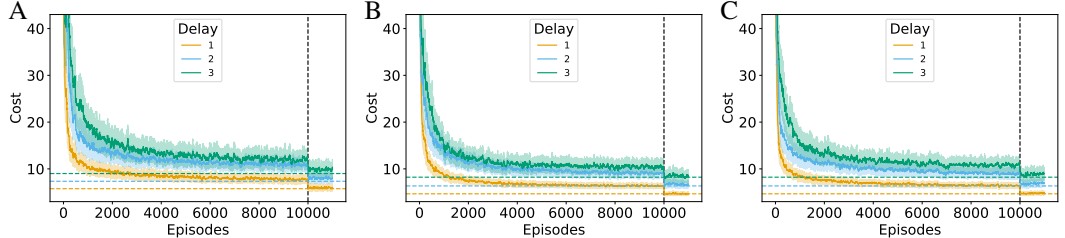

Figure 5: **Closed-loop Bio-OFC converges to optimally achievable cost given the delay.** The solid (shaded) curve depicts the mean ($\pm$SEM) of 20 runs with different random initial weights, smoothed using a running median with window size 51. The dashed horizontal lines show the asymptotic values from Fig. 3. The dashed vertical line depicts the time when learning was stopped and the exploratory stochasticity in the controller removed. Cost as function of episodes, using plasticity rules Eqs. (13-16) for the filter and Eqs. (21-23) for the controller simultaneously, for **(A)** 2-d observations (LDS1, see text for details), **(B)** 3-d observations (LDS2) and **(C)** 3-d observations (LDS2) with an over-representing Bio-OFC that assumes 3 instead of the actual 2 latent dimensions.

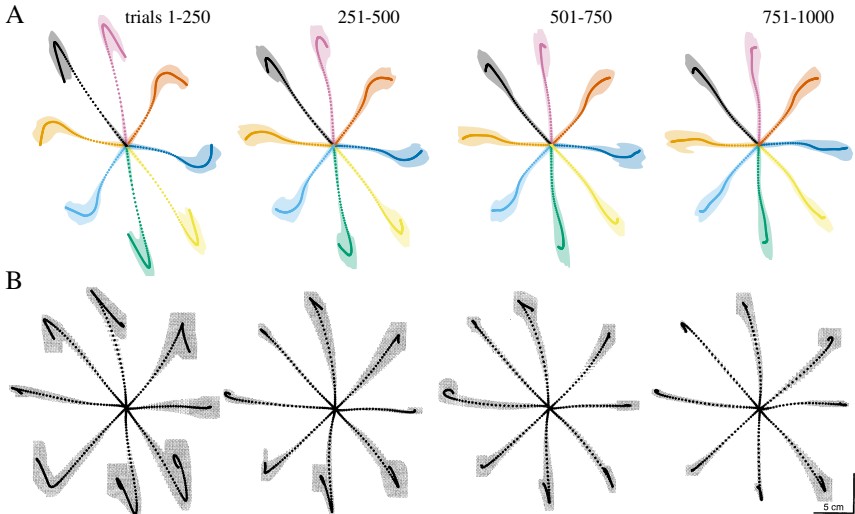

Figure 6: **Bio-OFC captures human performance when learning to adapt to a force field. (A)** Model trajectories during training. Performance plotted during the first, second, third, and final 250 targets. Dots show the mean and are 10 ms apart, shaded area shows a kernel density estimate thresholded at 0.04. **(B)** Averages$\pm$SD of human hand trajectories during training [1]. Copyright ©1994 Society for Neuroscience.

reveals that the network converged to the optimal cost. Different noise levels $\sigma$ and momenta $m$ are considered in Supplementary Figs. S8 and S9 respectively.

## 4.2  Reaching movements

To connect back to a biological sensory-motor control task, we considered the task of making reaching movements in the presence of externally imposed forces from a mechanical environment [1] (Supplementary Material). We initialized the weights of our network to the values that are optimal in a null force field, using a unit time of 10 ms and a sensory delay of 50 ms (i.e. $\tau = 5$), as has been measured experimentally [30]. Bio-OFC successfully captures the characteristics of human trajectories in the null field as well as the force field, cf. Fig. S10. Bio-OFC adapts to the force field by updating its weights according to plasticity rules Eqs. (13-16) for the filter and Eqs. (21-23) for the controller. Figs. 6 and S11 show that this captures human performance during the training period. Switching off learning in the controller yields virtually identical results, cf. Fig. S12, thus learning is driven primarily by changes in the estimator. Using signal-dependent motor noise in the plant [31], which increases with the magnitude of the control signal, also yields similar results, cf. Figs. S13-S14.

### 4.3 Simplified winged flight

For our final example, we designed an OpenAI gym [32] environment which simulates winged flight in 2-d with simplified dynamics (cf. Fig. S15). Here, the agent controls the flapping frequency of each wing individually, producing an impulse up and away from the wing (i.e. direction up and left when flapping the right wing). The agent receives sensory stimuli which are delayed by $100\,$ms (equivalent to $\tau = 5$ time-steps of the simulation). The goal of the agent is to fly to a fixed target and stabilize itself against gravity, the environment wind, and stochastic noise in the control system. The agent suffers a cost that is proportional to the L1 distance to the target, and the L1 magnitude of the control variables. This L1 cost was chosen to verify the flexibility of our algorithm when the cost deviates from the assumptions of LQR, where the cost is quadratic. We compare the performance of Bio-OFC to policy gradient. We find that, because of the delay, the agent trained with policy gradient overshoots the target and needs to backtrack, cf. Fig. S16a. However, the agent trained with Bio-OFC flies directly towards the target with no significant overshoot, cf. Fig. S16b. For more details and a video demonstration (`gym-fly-demo.mp4`) see the Supplementary Material.

## 5  Discussion

In this work, we developed a biologically plausible neural algorithm for sensory-motor control that learns a model of the world, and uses the ability to preview future, through a neurally plausible version of the Kalman filter, to learn an appropriate control policy. This neural circuit has the capacity to build an adequate representation of the appropriate state space, can deal with sensory delays, and actively explores the action space to execute appropriate control strategies.

We used a model-free controller, primarily due to its simplicity and biological plausibility. A model-based controller would need access to the model, i.e. weight matrices $A$ and $B$, which can result in a weight transport problem. However, model-based control has advantages such as higher sample efficiency and the ability to be transferable to other goals and tasks. An interesting question for future work would be how to combine state estimation via the Kalman filter with model-based control in a biologically plausible manner.

One limitation of this work is that it is in the framework of linear control theory. Locally linearized dynamics [33] has been suggested to generalize the Kalman filter. The inputs could also be processed using additional neural network layers to obtain a representation that renders the dynamics linear [34]. In several normative approaches towards neurally plausible representation learning, simply constraining neural activity to be nonnegative while retaining the same objective functions, allowed one to move from, say, PCA [35] to clusters [36] and manifolds [37]. Our work could be the starting point of a similar generalization.

We considered a uniform delay for all stimuli. In the case of motor control, proprioceptive feedback is faster than visual feedback [30]. Our model readily extends to the case of various, but known, delays for different modalities. The prediction error coding neurons merely need to combine predictions and measurements adequately, i.e. the synaptic delay associated with prediction has to match with the sensory delay for that modality. We believe learning the appropriate delay could be implemented by extending the state space using lag vectors [2], which we leave for future work.

In line with overall brain architecture [38] and the predictive coding framework [10], our model suggests the brain employs a recurrent network to generate predictions and actions by constantly attempting to match incoming sensory inputs with top-down predictions [11]. Our model can also be mapped to brain regions putatively contributing to optimal feedback control [30, 39]. Specifically, it has been proposed that the cerebellum performs system identification, parietal cortex performs state estimation, and primary and premotor cortices implement the optimal control policy by transforming state estimates into motor commands. Also, basal ganglia may be involved in processing cost/reward [39, 40].

This work proposed a concrete neural architecture that takes up the challenge of online control in a changing world [41], with delay, and using biologically plausible synaptic update rules. The learning algorithm performs well for several tasks, even with some drastic approximations. Future exploration of its limitations would provide further insights into the general nature of biologically constrained control systems.

## Acknowledgments and Disclosure of Funding

AS thanks A. Acharya, S. Saha and S. Sridharan for discussions. JF, SG, SF and DC were internally funded by the Simons Foundation. AS was partly funded by Simons Foundation Neuroscience grant SF 626323 during this work.

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
