# Neural optimal feedback control
# with local learning rules
# – Supplementary Material –

**Johannes Friedrich**[1]  **Siavash Golkar**[1]  **Shiva Farashahi**[1]

**Alexander Genkin**[5]  **Anirvan M. Sengupta**[2,3,4]  **Dmitri B. Chklovskii**[1,5]

[1] Center for Computational Neuroscience, Flatiron Institute
[2] Center for Computational Mathematics, Flatiron Institute
[3] Center for Computational Quantum Physics, Flatiron Institute
[4] Department of Physics and Astronomy, Rutgers University
[5] Neuroscience Institute, NYU Medical Center

{jfriedrich,sgolkar,sfarashahi,dchklovskii}@flatironinstitute.org
{alexander.genkin,anirvans.physics}@gmail.com

## 1  Linear-quadratic regulator (LQR)

The optimal control problem is to determine an output feedback law that minimizes the expected value of a cost criterion. If the cost $J$ is quadratic, the optimal output feedback is a linear control law known as linear-quadratic regulator (LQR).

$$J = \boldsymbol{x}_T^\top \boldsymbol{Q} \boldsymbol{x}_T + \sum_{t=0}^{T-1} \left( \boldsymbol{x}_t^\top \boldsymbol{Q} \boldsymbol{x}_t + \boldsymbol{u}_t^\top \boldsymbol{R} \boldsymbol{u}_t \right) \tag{S1}$$

$$\boldsymbol{u}_t = -\boldsymbol{K}_t \boldsymbol{x}_t \quad \text{with control gain} \quad \boldsymbol{K}_t = (\boldsymbol{B}^\top \boldsymbol{P}_{t+1} \boldsymbol{B} + \boldsymbol{R})^{-1} \boldsymbol{B}^\top \boldsymbol{P}_{t+1} \boldsymbol{A} \tag{S2}$$

where $\boldsymbol{P}_t$ is determined by the dynamic Riccati equation that runs backwards in time

$$\boldsymbol{P}_{t-1} = \boldsymbol{A}^\top \boldsymbol{P}_t \boldsymbol{A} - \left( \boldsymbol{A}^\top \boldsymbol{P}_t \boldsymbol{B} \right) \left( \boldsymbol{R} + \boldsymbol{B}^\top \boldsymbol{P}_t \boldsymbol{B} \right)^{-1} \left( \boldsymbol{B}^\top \boldsymbol{P}_t \boldsymbol{A} \right) + \boldsymbol{Q} \tag{S3}$$

from terminal condition $\boldsymbol{P}_T = \boldsymbol{Q}$. Linear-quadratic-Gaussian (LQG) control uses the Kalman estimate $\hat{\boldsymbol{x}}$ in the controller, $\boldsymbol{u}_t = -\boldsymbol{K}_t \hat{\boldsymbol{x}}_t$.

## 2  Experimental details

The experiments to produce the figures of the paper were performed on a Linux-based (CentOS) desktop with Intel Xeon CPU E5-2643 v4 @ 3.40GHz (6 cores) and 128 GB of RAM. No usage of a GPU was made. We used SciPy's [1] minimize function (with the default BFGS algorithm) to optimize the learning rate for Fig. 2 (which took 98 s) and the optimal gains $\boldsymbol{K}$ and $\boldsymbol{L}$ for Fig. 3 (which, dependent on the delay, took from few seconds up to 2 minutes).

To produce Figs. 4 and 5 (also supporting Figs. S6-S9), 20 parallel runs with different random seeds for 10000+1000 episodes took 9.3 s for open loop and 14.0 s for closed loop control, largely irrespective of the considered LDS and delay. The learning rates used in those experiments were obtained earlier with Optuna [2]. This hyperparameter optimization was performed on a linux-based (CentOS) cluster with Intel Xeon CPU E5-2680 v4 @ 2.40GHz (14 cores) and 512 GB of RAM, dedicating an individual node to each combinatorial choice of LDS, control setting (open/closed

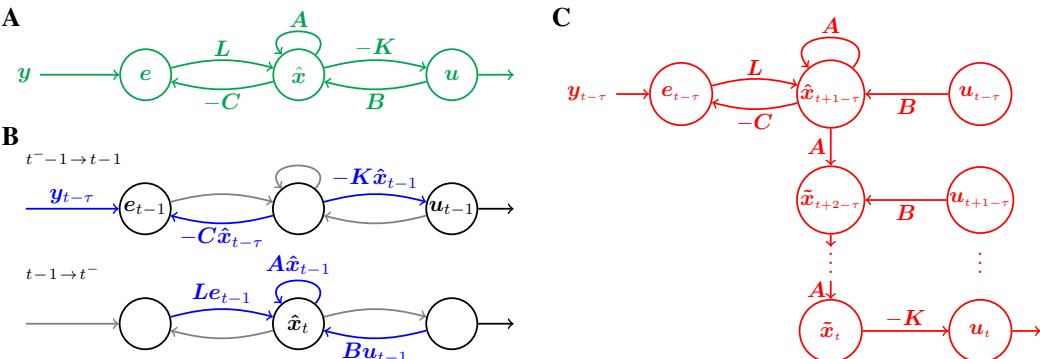

Figure S1: **Schematic of our proposed neural networks for OFC and alternative artificial neural network. (A)** Schematic of our proposed neural network (Bio-OFC). Nodes are annotated with the quantity they represent in their firing rates; edges are annotated with synaptic weights. **(B)** Input currents when updating $e_{t-1}$, $u_{t-1}$ (top), and $\hat{x}_t$ (bottom). $\hat{x}_t$ is updated directly using the delayed measurement $y_{t-\tau}$, even for delays $\tau > 1$. **(C)** The alternative artificial neural network (ANN) that updates the past estimate $\hat{x}_{t+1-\tau}$, and predicts forward in time to estimate $\tilde{x}_{t+2-\tau}, ..., \tilde{x}_t$, would require biologically implausible weight copies of $A$ and $B$, as well as a memory of past controls $u_{t-\tau}, ..., u_{t-1}$.

loop), and delay. Using a computing budget of 1000 trials optimization took about 130 min for open loop and 190 min for closed loop control.

Computational complexity of our algorithms are the same as that of policy gradient and Kalman filtering. That is, our approximations and biological implementation do not change the computational complexity of these methods.

## 3 Code

Code to reproduce the figures in the paper can be found in the GitHub repository `https://github.com/j-friedrich/neuralOFC`.
Requirements: `python`, `matplotlib`, `numpy`, `scipy`
The hyperparameters obtained with optuna [2] are provided in the subdirectory `results`.
To recreate a figure run the corresponding script. Figures will be saved in the subdirectory `fig`.

## 4 Learning rules

The marginal mean-squared error decreases if the angle between the gradient $g_\theta$ for parameter $\theta$ and the update $\Delta\theta$ is less than $90°$, i.e. $g_\theta^\top \Delta\theta > 0$. Thus to obtain $\Delta\theta$ we can left-multiply the gradient $g_\theta$ by any real square matrix $M$ that is positive definite, i.e. its symmetric part $\frac{1}{2}(M + M^\top)$ has positive real eigenvalues, thus that $g_\theta^\top(M + M^\top)g_\theta > 0$ while the anti-symmetric part always satisfies $g_\theta^\top(M - M^\top)g_\theta = 0$. Replacing $C^\top$ with $L$ to obtain the learning rules, Eqs. (13-14), corresponds to multiplication of the gradients, Eqs. (11,12), by the product of $L$ and the Moore-Penrose inverse of $C^\top$, $M = LC^{\top+}$. We therefore initialize $C$ and $L$ in such a way that $LC^{\top+}$ is positive definite (but not necessarily symmetric).

While we can initialize this way, a further issue is whether the learning rules still minimize the objective at the end of training upon convergence, i.e. near the optimum given by the Kalman filter. Using Eqs. (11-14), we have $g_\theta^\top \Delta\theta = v_{t-\tau}e_t^\top CLe_tv_{t-\tau}^\top$, where $v \in \{\hat{x}, u, e\}$ for $\theta \in \{A, B, L\}$. Hence it suffices if either $CL$ or $LC^{\top+}$ is positive definite. (If the matrices $C$ and $L$ are not square, one of the products won't have full rank, and have eigenvalues that are zero.) For the Kalman filter holds $L = A\Sigma C^\top(C\Sigma C^\top + W)^{-1}$, cf. Eq. (7). It follows that $CL = CA\Sigma C^\top(C\Sigma C^\top + W)^{-1}$. It is reasonable to assume this is positive semi-definite: For now assume $A = I$, then $CA\Sigma C^\top$ is the covariance due to uncertainty of the state, and $(C\Sigma C^\top + W)$ is the total covariance due to uncertainty of the state plus observation noise. The product $CL$ can be considered as a ratio of these

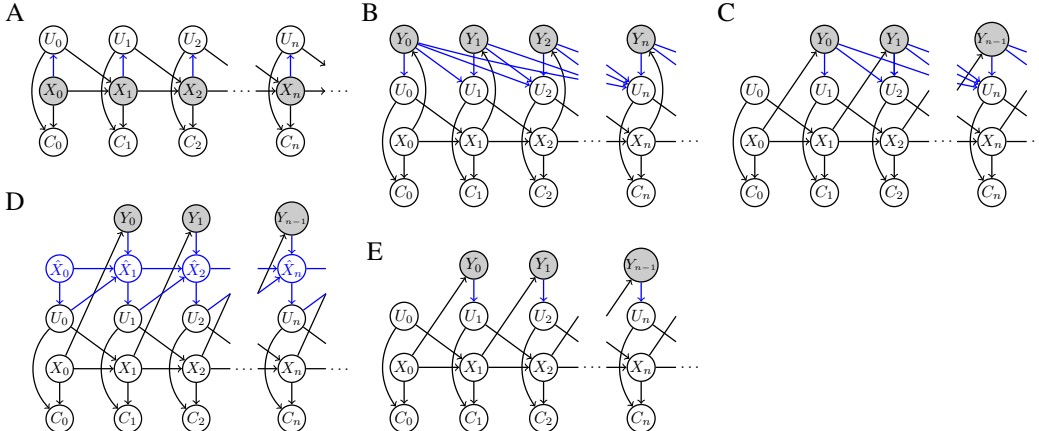

Figure S2: **Probabilistic graphical models of MDPs and POMDPs.** Grey colored nodes are observed, black and blue colored edges depend on the environment and actor, respectively. **(A)** MDP; the state is observed directly and the optimal action $u_t$ depends only on the current state $x_t$. **(B)** POMDP; the state is only partially observable and the action $u_t$ depends on the history of observations $y_0, ..., y_t$. **(C)** POMDP with delayed observations (here $\tau = 1$); the action $u_t$ depends on the limited history of available observations $y_0, ..., y_{t-\tau}$. **(D)** Model-based controller that produces action $u_t$ based on the current state $\hat{x}_t$ of the internal model that effectively summarizes past observations. Our Bio-OFC is an instance of such a controller. **(E)** Model-free memory-less controller that produces action $u_t$ based on the currently available delayed observation $y_{t-\tau}$.

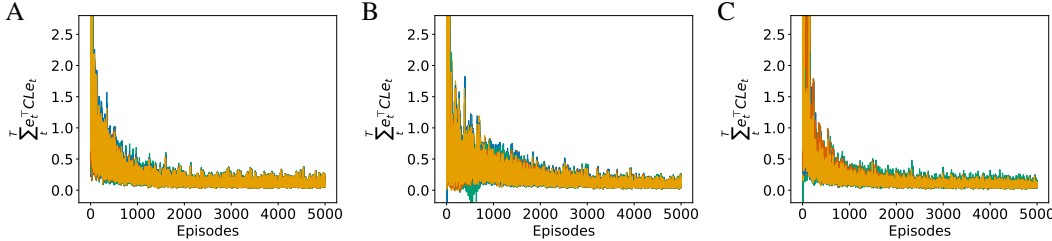

Figure S3: **Alignment of weight update $\Delta\boldsymbol{\theta}$ and gradient $g_{\boldsymbol{\theta}}$.** Average $\sum_t^T e_t^\top \boldsymbol{CL}e_t$ over one episode during system identification as function of episodes for **(A)** 2-d observations (LDS1), **(B)** 3-d observations (LDS2) and **(C)** 3-d observations (LDS2) with an over-representing Bio-OFC that assumes 3 instead of the actual 2 latent dimensions. All 20 runs are shown using different colors.

covariances, which is close to $\boldsymbol{I}$ for small observation noise. Our learning rules scale the gradients by these covariances. Because $\boldsymbol{A} = \boldsymbol{I} + \mathcal{O}(\Delta t)$ the product $\boldsymbol{CL}$ remains positive definite, as long as the time discretization is not too coarse. Indeed, in the continuous limit of the Kalman-Bucy filter the Kalman gain is simply $\boldsymbol{L} = \boldsymbol{\Sigma C}^\top \boldsymbol{W}^{-1}$.

$\boldsymbol{g}_{\boldsymbol{\theta}}^\top \Delta\boldsymbol{\theta} > 0$ holds at the end and beginning of training, the latter due to the way we initialize $\boldsymbol{C}$ and $\boldsymbol{L}$. However, what happens throughout learning? When performing stochastic gradient descent only the average update aligns with the negative gradient, whereas individual updates could even increase the objective. Similarly $\boldsymbol{g}_{\boldsymbol{\theta}}^\top \Delta\boldsymbol{\theta} > 0$ has to hold only on average. Note that $\boldsymbol{g}_{\boldsymbol{\theta}}^\top \Delta\boldsymbol{\theta} = \boldsymbol{v}_{t-\tau} e_t^\top \boldsymbol{CL}e_t \boldsymbol{v}_{t-\tau}^\top$, where $\boldsymbol{v} \in \{\hat{\boldsymbol{x}}, \boldsymbol{u}, \boldsymbol{e}\}$ for $\boldsymbol{\theta} \in \{\boldsymbol{A}, \boldsymbol{B}, \boldsymbol{L}\}$. We therefore revisited the simulations of Fig. 4 for delay=1 and kept track of $e_t^\top \boldsymbol{CL}e_t$, which needs to be positive on average. While $e_t^\top \boldsymbol{CL}e_t$ was negative for 7.1% of the individual updates for LDS2 (0% for LDS1), the average over one episode $\sum_t^T e_t^\top \boldsymbol{CL}e_t$ was negative for merely 0.036% of the episodes, cf. Fig. S3, and always positive if averaged over multiple episodes. Although we do not present a theoretical derivation to show that $\mathbb{E}[e_t^\top \boldsymbol{CL}e_t] > 0$ the simulations show this is the case.

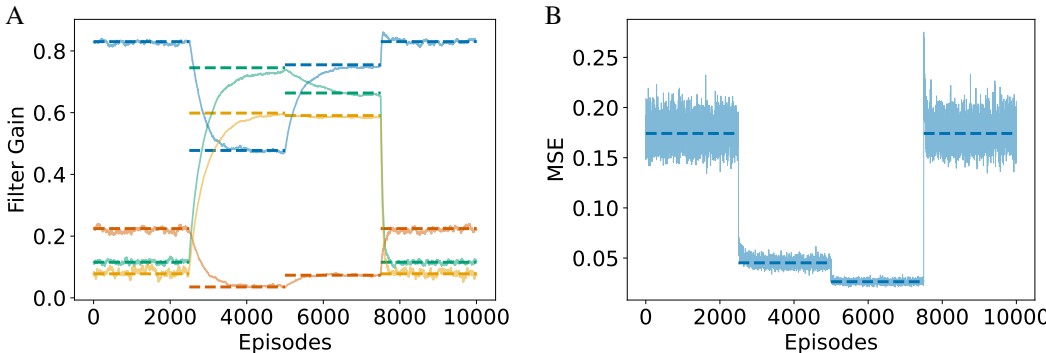

Figure S4: **Adaptive filtering under varying noise levels.** Analogous plots to Fig. 2, but using the non-local SGD learning rule (12) instead of Eq. (14) which replaces $C^\top$ with $L$ to render the learning rule local.

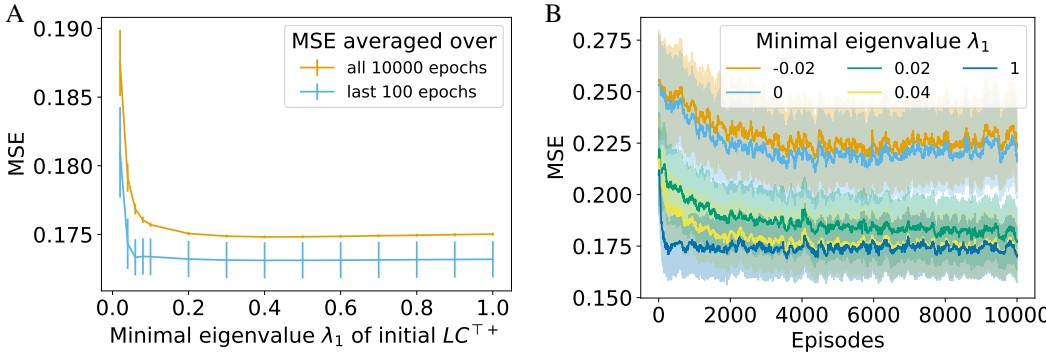

Figure S5: **Dependence on initial alignment of $C^\top$ and $L$.** (A) Average MSE ($\pm$SEM) when the minimal eigenvalue $\lambda_1$ of $LC^{\top+}$ at initialization is varied while the maximal eigenvalue is constant, $\lambda_2 = 1$. Performance does not change in a statistically significant way over a wide range of initial alignments. (B) Convergence to optimal asymptotic performance occurs if $\lambda_1 > 0$ but not if $\lambda_1 \leq 0$.

## 5 Consequences of replacing $\mathrm{C}^\top$ with $\mathrm{L}$

To obtain local learning rules, Eqs. (13-16), we replaced $C^\top$ in the gradient formulas, Eqs. (11-12), with $L$. Fig. S4 repeats the experiment of Fig. 2 without replacing $C^\top$. The learning rate that minimizes the average MSE over all episodes is smaller, convergence slower, and the average MSE even marginally larger than in Fig. 2. However, the overshooting after the covariance change at 2500 episodes (in Fig. 2A) does not occur. Thus we find that the replacement does not harm performance.

We further noted that replacing $C^\top$ with $L$ corresponds to multiplication of the gradients by $LC^{\top+}$, and we therefore initialize $C$ and $L$ in such a way that $LC^{\top+}$ is positive definite, i.e. its symmetric part has positive real eigenvalues. We investigated the dependence on initial alignment of $C^\top$ and $L$ for the LDS1 experiment and found that the performance does not change in a statistically significant way over a wide range of initial alignments that we considered. In more detail, we learned the Kalman gain $L$ for LDS1 ($C = I$), initializing $L$ as $\begin{pmatrix} 1 - a & a \\ a & 1 - a \end{pmatrix}$, which has eigenvalues $\lambda_1 = 1 - 2a$ and $\lambda_2 = 1$. If $a = 0$ then $L$ and $C^\top$ are perfectly aligned, for $a = 0.5$ eigenvalue $\lambda_1$ of $L$, and thus of the symmetric part of $LC^{\top+}$, becomes zero. The asymptotic performance (operationally defined as average MSE of the last 100 episodes) for $\lambda_1 > 0.02$ did not differ in a statistical significant way (p>0.6, two sided t-test), but convergence was slower for small but positive $\lambda_1 \gtrsim 0$, cf. Fig. S5. Panel B shows that for $\lambda_1 \leq 0$ performance did not converge to the optimal asymptotic value. For each value of $\lambda_1$ the learning rate was tuned to minimize the average MSE over all episodes.

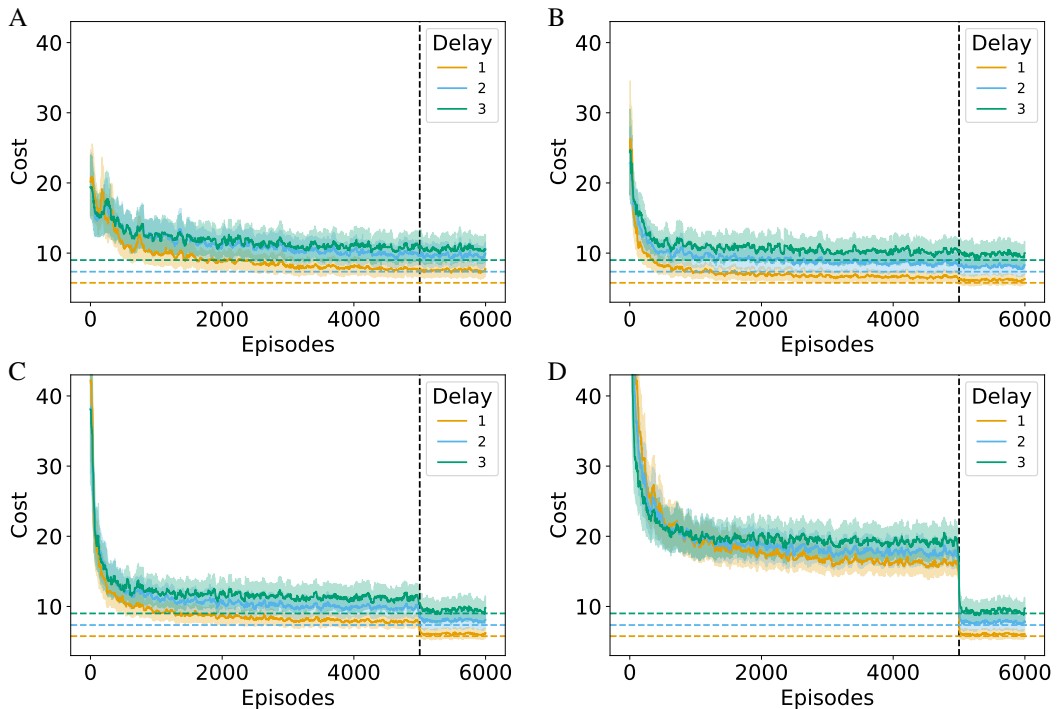

Figure S6: **Open-loop training of Bio-OFC on LDS1 for different controller noise levels.** Cost as function of episodes for **(A)** $\sigma = 0.05$, **(B)** $\sigma = 0.1$, **(C)** $\sigma = 0.2$, and **(D)** $\sigma = 0.5$, cf. Fig. 5 for details. Panel C is identical to Fig. 4D.

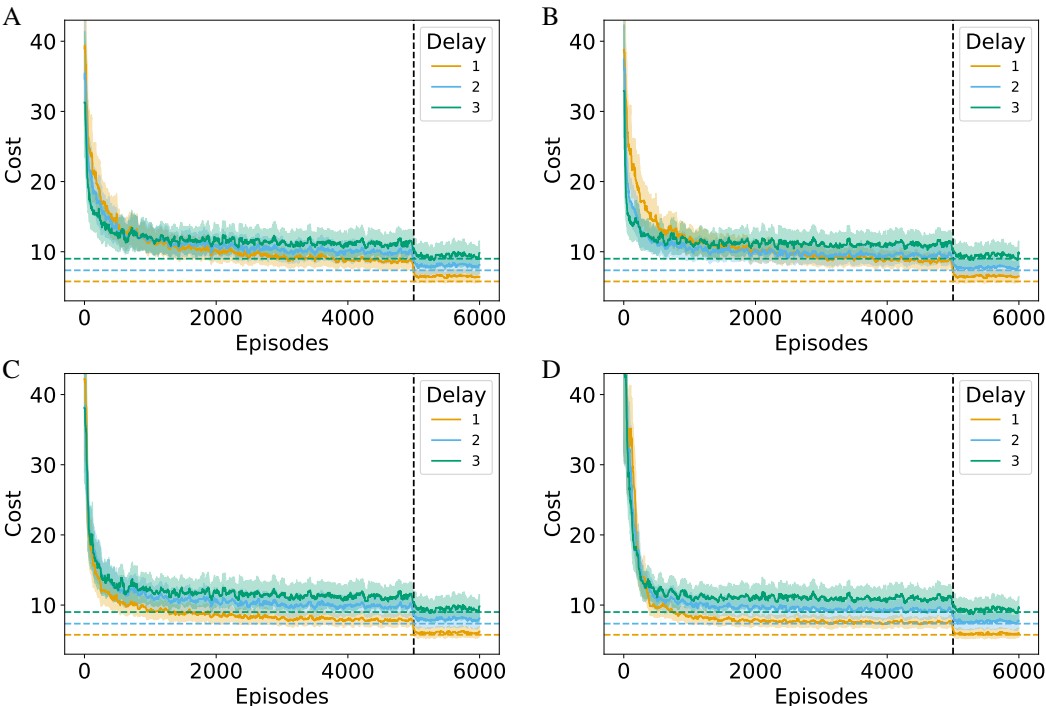

Figure S7: **Open-loop training of Bio-OFC on LDS1 using different momenta in the controller update.** Cost as function of episodes for **(A)** $m = 0$, **(B)** $m = 0.9$, **(C)** $m = 0.99$, and **(D)** $m = 0.9995$, cf. Fig. 4 for details. Panel C is identical to Fig. 4D.

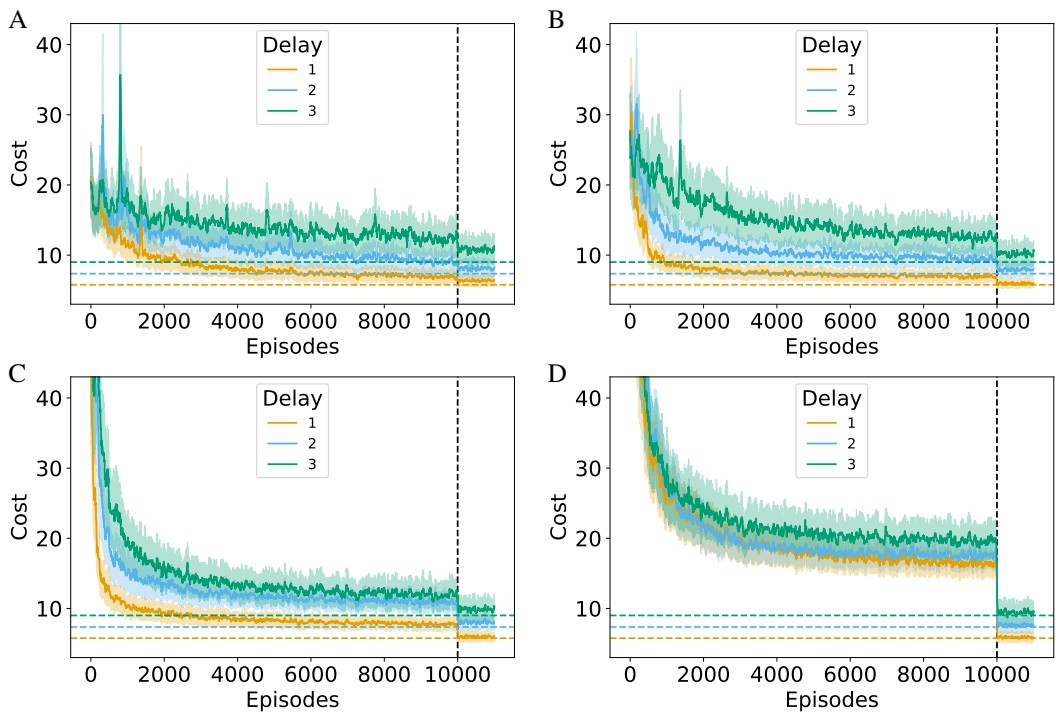

Figure S8: **Closed-loop training of Bio-OFC on LDS1 for different controller noise levels.** Cost as function of episodes for **(A)** $\sigma = 0.05$, **(B)** $\sigma = 0.1$, **(C)** $\sigma = 0.2$, and **(D)** $\sigma = 0.5$, cf. Fig. 5 for details. Panel C is identical to Fig. 5A.

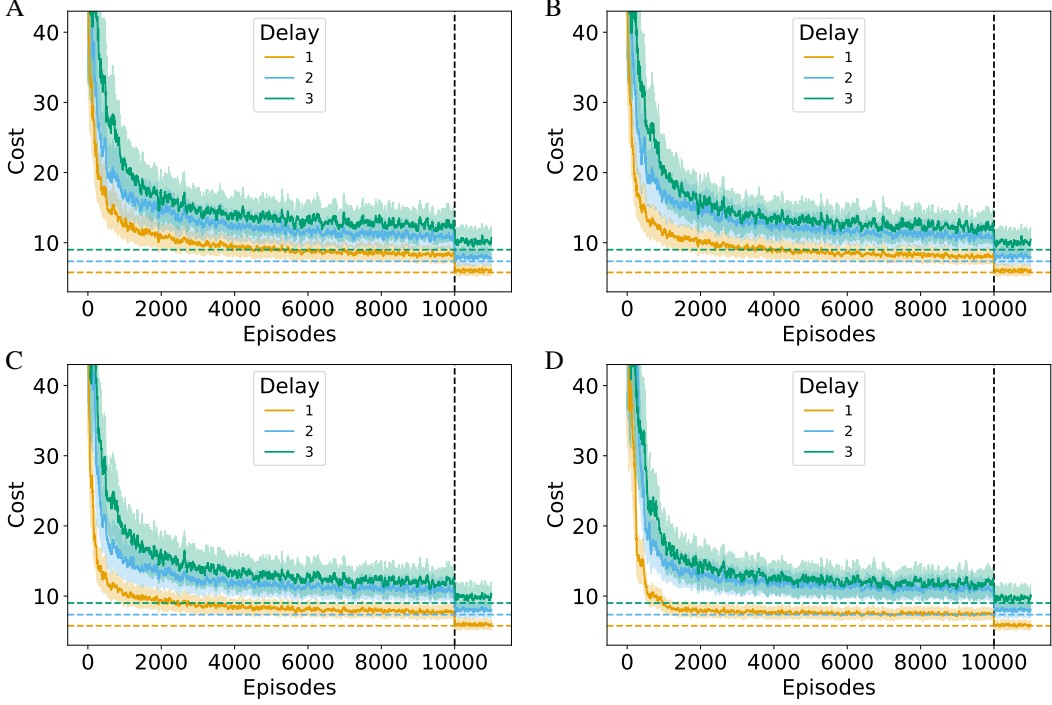

Figure S9: **Closed-loop training of Bio-OFC on LDS1 using different momenta in the controller update.** Cost as function of episodes for **(A)** $m = 0$, **(B)** $m = 0.9$, **(C)** $m = 0.99$, and **(D)** $m = 0.9995$, cf. Fig. 5 for details. Panel C is identical to Fig. 5A.

# 6 Learning of a sensory-motor control task

We considered the task of making reaching movements in the presence of externally imposed forces from a mechanical environment [3]. The movements are restricted to a fixed $z$-plane and the 6-dimensional state vector contains the position, velocity and acceleration in $x$ and $y$ direction, $\boldsymbol{x} = (p_x, p_y, v_x, v_y, a_x, a_y)^\top$. In the absence of a force field the system is described by matrices

$$\boldsymbol{A} = \begin{pmatrix} 1 & 0 & 1 & 0 & 0 & 0 \\ 0 & 1 & 0 & 1 & 0 & 0 \\ 0 & 0 & 1 & 0 & 1 & 0 \\ 0 & 0 & 0 & 1 & 0 & 1 \\ 0 & 0 & 0 & 0 & 1 & 0 \\ 0 & 0 & 0 & 0 & 0 & 1 \end{pmatrix}, \quad \boldsymbol{B} = \begin{pmatrix} 0 & 0 \\ 0 & 0 \\ 0 & 0 \\ 0 & 0 \\ 1 & 0 \\ 0 & 1 \end{pmatrix}, \quad \boldsymbol{C} = \boldsymbol{I} \tag{S4}$$

For simplicity we follow [4] and assume that the action $\boldsymbol{u}$ is already in Cartesian coordinates as opposed to controlling the torques applied on joints. We leave it for future work to further tighten the connection to biological motor control. We assumed time units of 10 ms, length units of 1 cm and – in line with experimental data [5] – a measurement delay of 50 ms. Further, the noise covariances and reward matrices were parametrized as

$$\boldsymbol{V} = v \operatorname{diag}(1, 1, .1, .1, .01, .01), \quad \boldsymbol{W} = \boldsymbol{V}, \quad \boldsymbol{Q} = \operatorname{diag}(q_1, q_1, q_2, q_2, 0, 0), \quad \boldsymbol{R} = \boldsymbol{I} \tag{S5}$$

where the different scales in $\boldsymbol{V}$ reflect the different numerical scales of position, velocity and acceleration. The forces were computed as a function of the velocity. Application of the force field changes $\boldsymbol{A}$ to

$$\boldsymbol{A} \leftarrow \boldsymbol{A} + f \begin{pmatrix} 0 & 0 & 0 & 0 & 0 & 0 \\ 0 & 0 & 0 & 0 & 0 & 0 \\ 0 & 0 & -10.1 & -11.2 & 0 & 0 \\ 0 & 0 & -11.2 & 11.1 & 0 & 0 \\ 0 & 0 & 0 & 0 & 0 & 0 \\ 0 & 0 & 0 & 0 & 0 & 0 \end{pmatrix} \tag{S6}$$

where the numerical values of the non-zero matrix entries were taken from [3]. We chose parameters $v, q_1, q_2, f$ that result in a qualitative match with the experimental data, see Fig. S10 ($v = 10^{-4}$, $q_1 = 10^{-5}$, $q_2 = 0.002$, $f = 0.002$). The optimal filter gain was determined by minimizing the mean squared prediction error over 1000 trajectories. This demonstrated that our network, that approximates OFC, can adequately describe experimental behavior, which is mostly a testimony to the success of OFC. It is standard in linear control theory to put the target state at the origin. This can be achieved by using variables related to the difference between the initial state and the target state. As a result, we can use the same estimator/controller for each reach condition and the different reach conditions correspond to different initial states $x_0$. The more interesting question is whether our plasticity rules Eqs. (13-16) for the filter and Eqs. (21-23) for the controller, capture human performance during the training period. Fig. 6 shows that this is indeed the case and after about 1000 episodes the trajectories are close to straight lines. Fig. S12 repeats this analysis but updates only the system matrices $\boldsymbol{A}, \boldsymbol{B}, \boldsymbol{C}$ and Kalman gain $\boldsymbol{L}$, while keeping the controller weights $\boldsymbol{K}$ fixed. Even switching off learning in the controller yields similar results, thus learning is driven primarily by changes in the estimator. Because the force field alters the system, greater changes in the part that performs system identification, i.e. the estimator, are somewhat to be expected.

To strengthen the connection to biological motor control we reran the reaching task using signal-dependent motor noise in the plant [6] which increases with the magnitude of the control signal. We scaled the amount of noise by the norm of $\boldsymbol{u}$, i.e. we replaced the dynamics $\boldsymbol{x}_{t+1} = \boldsymbol{A}\boldsymbol{x}_t + \boldsymbol{B}\boldsymbol{u}_t + \boldsymbol{v}_t$ with $\boldsymbol{x}_{t+1} = \boldsymbol{A}\boldsymbol{x}_t + \boldsymbol{B}\boldsymbol{u}_t + |\boldsymbol{u}_t|\boldsymbol{v}_t$. The results, shown in Figs. S13 and S14, are similar to the earlier results obtained with additive noise (Figs. S10 and 6).

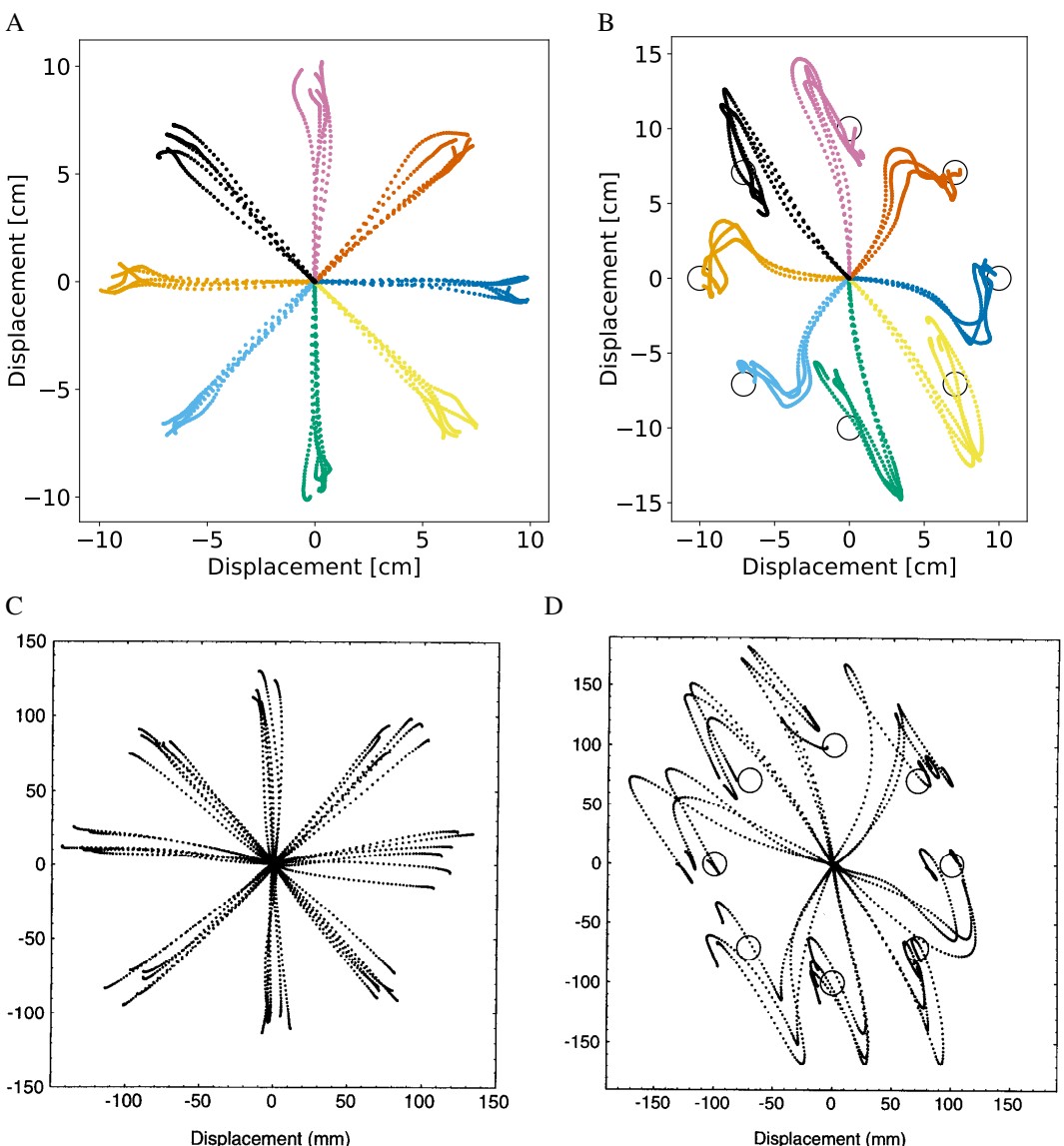

Figure S10: **Optimal feedback control qualitatively captures hand reaching trajectories.** Model trajectories **(A)** in a null force field and **(B)** during initial exposure to a force field. Typical human trajectories **(C)** in a null force field and **(D)** during initial exposure to a force field [3]. Copyright ©1994 Society for Neuroscience.

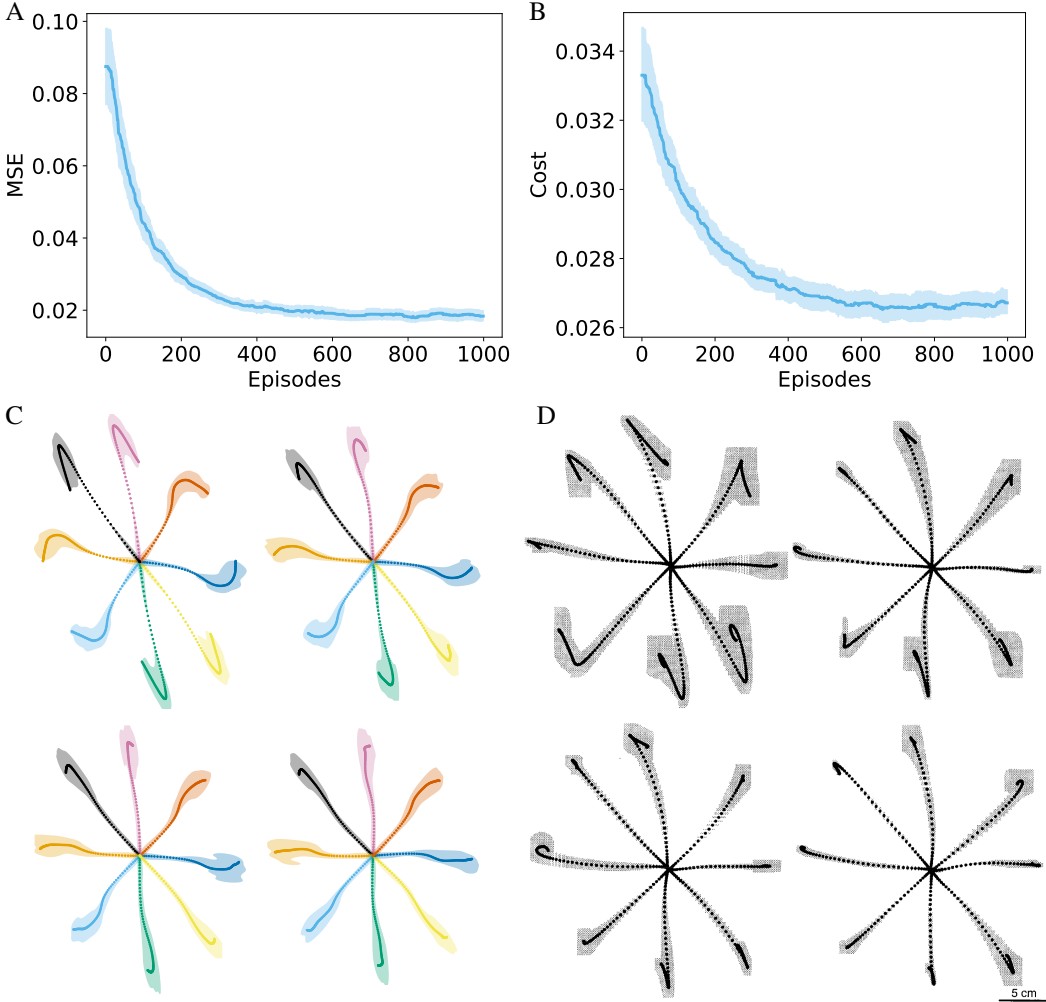

Figure S11: **Learning to adapt to a a force field.** **(A)** Mean squared prediction error and **(B)** cost during training. **(C)** Model trajectories during training. Performance plotted during the first (top left), second (top right), third (bottom left), and final (bottom right) 250 targets. Dots show the mean and are 10 ms apart, shaded area shows a kernel density estimate thresholded at 0.04. **(D)** Averages±SD of human hand trajectories during training [3]. Copyright ©1994 Society for Neuroscience.

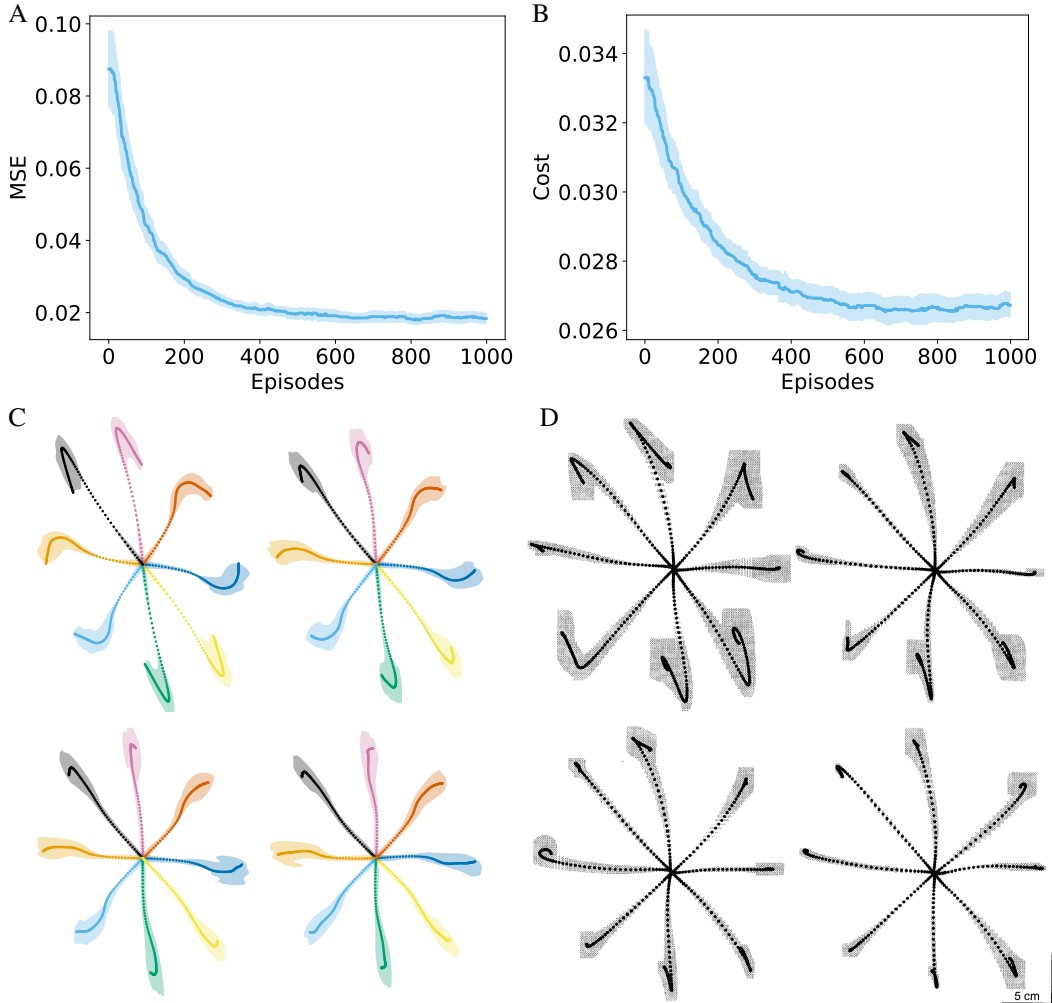

Figure S12: **Learning to adapt to a a force field without adaptation of controller weights. (A)** Mean squared prediction error and **(B)** cost during training. **(C)** Model trajectories during training. Performance plotted during the first (top left), second (top right), third (bottom left), and final (bottom right) 250 targets. Dots show the mean and are 10 ms apart, shaded area shows a kernel density estimate thresholded at 0.04. **(D)** Averages±SD of human hand trajectories during training [3]. Copyright ©1994 Society for Neuroscience.

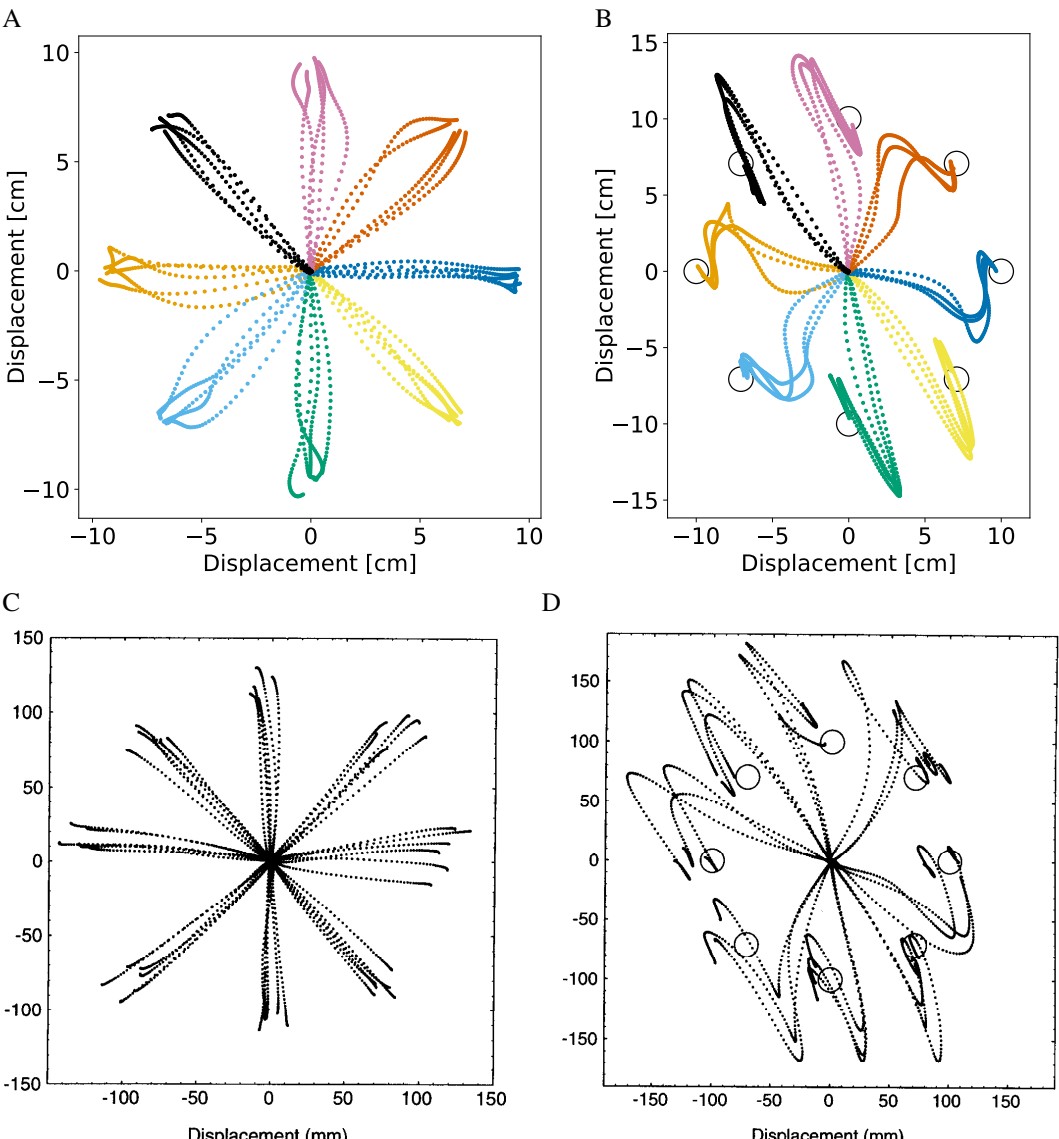

Figure S13: **Optimal feedback control with multiplicative signal-dependent noise qualitatively captures hand reaching trajectories.** Model trajectories **(A)** in a null force field and **(B)** during initial exposure to a force field. Typical human trajectories **(C)** in a null force field and **(D)** during initial exposure to a force field [3]. Copyright ©1994 Society for Neuroscience.

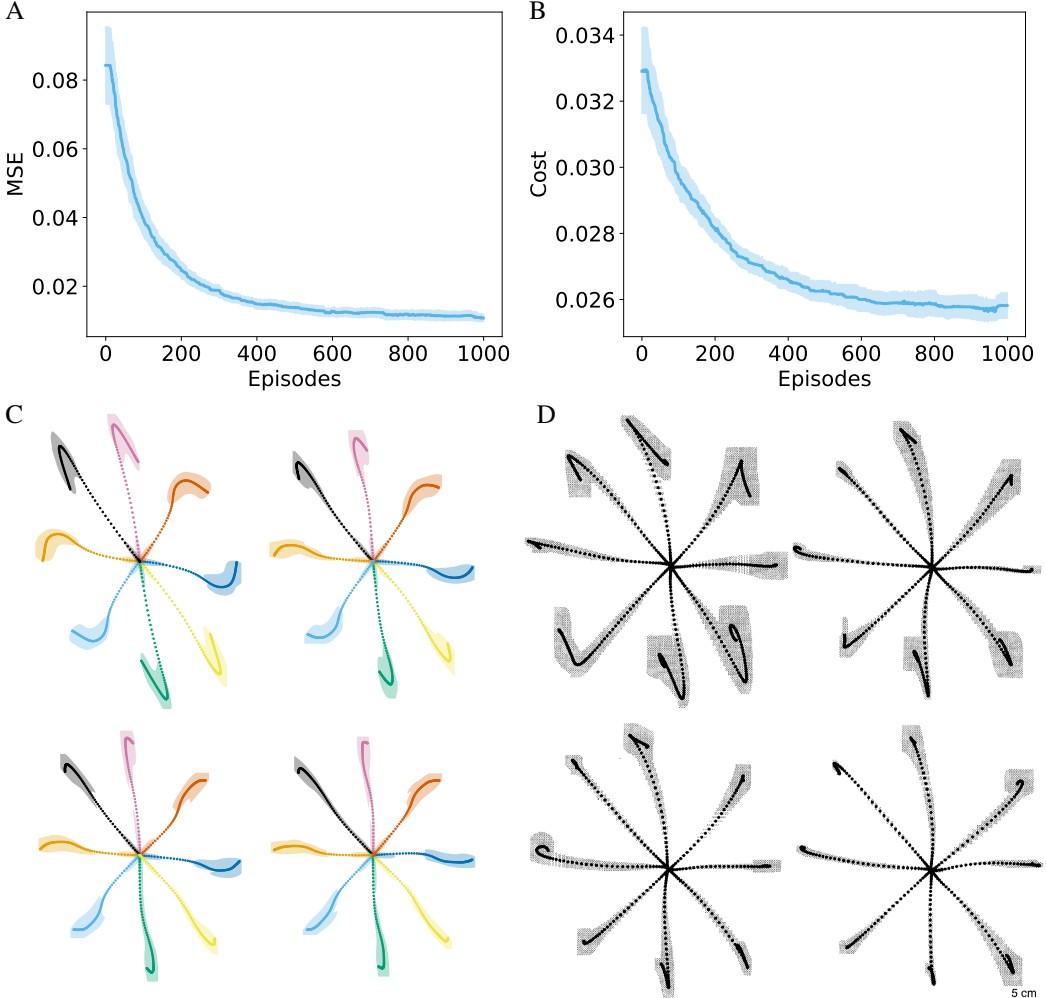

Figure S14: **Learning to adapt to a a force field with multiplicative signal-dependent control noise.** **(A)** Mean squared prediction error and **(B)** cost during training. **(C)** Model trajectories during training. Performance plotted during the first (top left), second (top right), third (bottom left), and final (bottom right) 250 targets. Dots show the mean and are 10 ms apart, shaded area shows a kernel density estimate thresholded at 0.04. **(D)** Averages±SD of human hand trajectories during training [3]. Copyright ©1994 Society for Neuroscience.

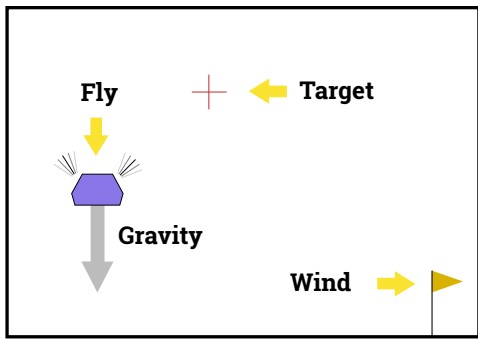

Figure S15: The fly simulation environment.

# 7 A simplified fly simulation environment

We designed Gym-Fly, an OpenAI gym [7] environment which simulates flying in a simplified 2-d environment (cf. Fig. S15). We implemented the physical simulations using the Box2D, a 2-d physics engine often used for games [8]. In this environment, the agent controls the flapping frequency of each wing individually. Each wing flap results in an impulse that is up and away from the wing (i.e. direction up and left when flapping the right wing) and for simplicity, we assume that the flapping frequency translates linearly to the magnitude of the impulse imparted at each time-step. Furthermore, we assume that negative flapping frequency results in an impulse in the opposite direction. While this is not realistic in many physical situations, it is necessary for a linear environment which can be described by Eq. (1). The environment has gravity, and wind that randomly increases or decreases at each time-step, up to a maximum value. Furthermore, the system suffers from stochastic noise in that the result of the agent's actions are corrupted before being implemented by the engine. The agent receives sensory stimuli that are composed of its 2-d location and 2-d velocity, as well as the measurement of the wind in the environment. However, these observations are delayed by $100\,\mathrm{ms}$, equivalent to $\tau = 5$ time-steps of the simulation.

The goal of the agent is to fly to a fixed target and stabilize itself against gravity, the environment wind, and stochastic noise in the system. The agent suffers a cost that is proportional to the distance to the target, and the magnitude of the control variables. To verify the flexibility of the Bio-OFC algorithm, we implemented the Gym-Fly environment to deviate from the assumptions used for deriving Bio-OFC in a number of ways. First, the cost suffered by the agent is not quadratic as required by LQR (cf. Eq. (S1)) but is an L1 distance. Second, the system noise that corrupts the control parameters was chosen to be uniform and not Gaussian.

Because of gravity, the description and control of the system with Eqs. (1) and (4), require a bias term. This is because the agent will fall if it stops flapping its wings, that is the point $\boldsymbol{x} = 0$ and $\boldsymbol{u} = 0$ is not a fixed point. Bio-OFC can be easily modified to allow for a bias term. The simplest way to derive the algorithm with a bias is to allow for a component of $\hat{\boldsymbol{x}}$ to be fixed at a constant value, i.e. by replacing $\hat{\boldsymbol{x}} \to (\hat{\boldsymbol{x}}, 1)$. In a biological setting this bias can be implemented as a thresholding mechanism in the post-synaptic neuron and does not violate the locality and biological plausibility of the learning rules.

We trained Bio-OFC in this environment in a closed-loop setting for a total of 30,000 episodes. The length of the episodes start at 150 time-steps and was linearly increased to 1000 time-steps at episode 500 after which they no longer increase. A video demonstration of how Bio-OFC learns to control this environment is given in `code-gym/gym-fly-demo.mp4`. However, the learning of the bias term, which controls the locations that the agent is stabilized, takes longer to be learned. This is understandable since deviations in the bias term lead to a constant cost at each time-step. However, failure to stabilize the agent against wind or gravity leads to a cost that will diverge in time. We also compared the performance of Bio-OFC to policy gradient. We find that, because of the delay, the agent trained with policy gradient overshoots the target and needs to backtrack, cf. Fig. S16a. However, the agent trained with Bio-OFC flies directly towards the target with no significant overshoot, cf. Fig. S16b.

The code for the Gym-Fly environment (`gym-fly/gym_fly/envs/fly_env.py`) as well as the detailed parameters of the training are provided in the accompanying code (`Bio-OFC gym-fly demo.ipynb`) in the GitHub repository `https://github.com/golkar/bio-ofc-gym`. Installation directions are given in `installation.txt`.

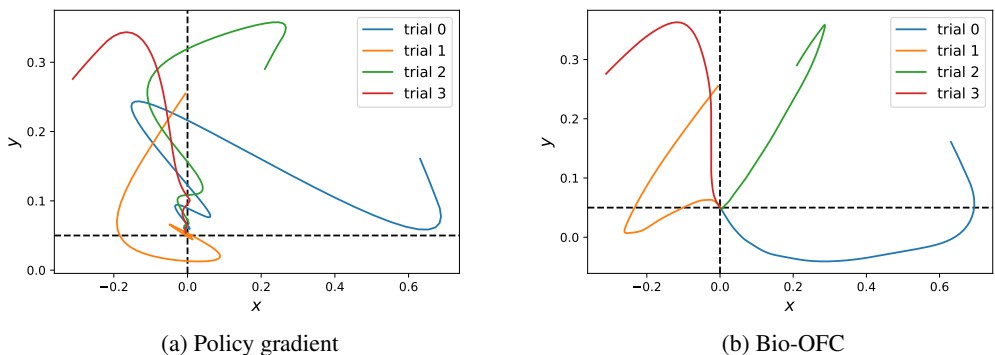

(a) Policy gradient                    (b) Bio-OFC

Figure S16: Flight trajectory of agents trained with policy gradient (left) and Bio-OFC (right) under the same initial conditions. The agent trained with policy gradient overshoots the target, whereas the agent trained with Bio-OFC has no significant overshoot.