# OpenReview forum: "Neural optimal feedback control with local learning rules"
_NeurIPS.cc/2021/Conference — NeurIPS 2021 Spotlight_

### Official Review · Reviewer_MZDW · 2021-07-12

**Rating:** 6
**Confidence:** 3

**Summary:**

The paper proposes a method (bio-OFC) for learning optimal feedback control policies in a neural network with local learning rules. The method builds on the dominant model of biological motor control, optimal feedback control (Todorov and Jordan, 2002), which, given knowledge of system dynamics and noise covariances, is able to account well for variance patterns in human motor behaviour. A major limitation of OFC as a model of biological motor systems is that it does not learn the system dynamics and previously has not clearly mapped on to biologically-plausible implementations.
In this work, the authors propose a method for adapting the OFC framework so that it instead learns a forward predictive model of the system dynamics, and uses policy gradients to learn control. Unlike the original OFC formulation, this method doesn’t require a priori knowledge of noise covariances. The authors perform experiments in three environments, a simple control task (discrete time double integrator), a classic psychophysical reaching task, and a ‘winged flight’ control task.


**Limitations And Societal Impact:**

Yes.

**Main Review:**

Overall the work provides a development on a previous model with similar aspirations to provide a neural implementation of OFC (Linkser, 2008), which although acknowledged in Table 1 with a checklist comparison of common vs different features to their own work, was unfortunately not simulated under the same tasks for a more in-depth comparison. This paper did however test their method more thoroughly than Linkser 2008 and on new tasks, which is valuable.
The authors claim that one of the novel contributions of their method is that it functions under delayed sensory feedback, however the original OFC implementation by Todorov and Jordan did consider the case of delayed sensory feedback. It is possible that learning under these delayed sensory conditions is more difficult to handle than in the 2002 control-only formulation, in which case the authors should clarify why this is so.
This paper compares the control cost performance of their method to optimal LQG control, a NN controller that is ‘biologically implausible’, and a controller which does not learn a predictive dynamics model and instead maps delayed sensory observations linearly to control output (Figure 3). The authors’ method appears to perform fairly well (Fig3, 4) as although it is expected that bio-OFC will not learn a solution quite as good as LQG control, its not very far off, and bio-OFC performs only a little bit worse than the ANN with weight copies (blue).
Overall the results section was a little hard to follow, I think because it wasn't immediately clear what was learned from the results from each experiment or why each experiment was run. Some important results were also relegated to Supplemental information, e.g. results from the reaching movement experiment, which would have been valuable to include in the main text, seeing as the link to biological motor control seems to be a major focus of this paper.
A way to strengthen the connection to biological motor control would be to run experiments in which there is signal-dependent motor noise in the plant (Harris & Wolpert, 1998), and compare behaviour to humans.


**Time Spent Reviewing:**

2.5

---

> ### Author Response · Authors · 2021-08-10
> **Response to reviewer MZDW**
>
> We thank the reviewer for their detailed review and constructive feedback.
>
> * **Comparison with Linsker.** While our submission has aspirations similar to Linsker's work, we actually achieve them without the shortcomings with regard to biological plausibility as described in Table 1. One crucial aspiration we had that goes beyond Linsker’s was to handle delayed feedback. Because Linsker’s model (as well as other previously proposed neural implementations) cannot handle delay it does not directly apply to our tasks.
>
> * **Comparison with Todorov and Jordan.** Todorov and Jordan did consider the case of delayed sensory feedback, but presented a computational-level theory and used non-biological control algorithms that do not map onto a neural implementation. We present such a neural implementation of OFC, which is a more difficult challenge due to the limiting constraints imposed by biology. Our comment in the introduction regarding the novelty of including delay was regarding previous work attempting biologically plausible system-ID and control. We will clarify that delay can be dealt with using the conventional OFC framework (i.e. Todorov and Jordan).
>
> * **Results section is hard to follow.** Due to space constraints we had to relegate some results to the Supplementary Material. Catering to the wider NeurIPS audience we included results in the main paper that could be of interest not only to the neuroscience but also the control community. If accepted, we will use the extra space to expand on the results section to make it more readable and include more of the reaching experiment results in the main text as requested.
>
>
> * **Strengthen connection to biological motor control by considering signal-dependent motor noise in the plant (Harris & Wolpert, 1998), and compare behaviour to humans.**
> This is a great suggestion. We reran the reaching task using signal-dependent noise. We scaled the amount of noise by the norm of $u$, i.e. we replaced the dynamics
> $x_{t+1} = A x_t + B u_t + v_t$ with $x_{t+1} = A x_t + B u_t + |u_t| v_t$. This did not change the results of Figures S7 and S8 qualitatively. We will include these preliminary simulations in the supplementary materials, and leave it for future work to further tighten the connection to biological motor control.

---

### Official Review · Reviewer_pVEa · 2021-07-17

**Rating:** 8
**Confidence:** 3

**Summary:**

The paper introduces a biologically plausible neural implementation of optimal feedback control, which considers that sensory feedback is delayed, and combines adaptive Kalman filtering and model free control. The proposed algorithm can be applied closed-loop, i.e., the algorithm can be used for online control such as natural neural networks, and does not need additional training and calibration phases. The performance of the method has been assessed in several simulation experiments. Simulation results confirm that appropriate sensory-motor control can be achieved.

The main contribution is the proposed network named Bio-OFC, which does not require information about noise and system dynamics and allows closed-loop control.

**Ethical Concerns:**

No ethical concerns. Simulation and synthetic data were used to assess the performance of the system.

**Limitations And Societal Impact:**

Major limitation has been addressed. Discussion of the impact on society is not yet relevant for this work.

**Main Review:**

The work is novel, timely, clearly written, well organised and well presented. The main contributions are clearly highlighted and motivated. The authors reasoning and rationales, and the use of methods is sound. Limitations have been discussed; code and supplemental material is provided. This is an interesting approach and useful feedback-based learning approach.

The work addresses a difficult task in a new way. The claims are supported by simulation studies. Computational complexity and run-times of the method are not discussed.


**Time Spent Reviewing:**

2.5

---

> ### Author Response · Authors · 2021-08-10
> **Response to reviewer pVEa**
>
> We thank the reviewer for their thoughtful and encouraging comments.
>
> **Computational complexity and run-times not discussed.** The run-times and hardware used are discussed in Sec. 2 of the supplementary materials. Computational complexity of our algorithms are the same as that of policy gradient and Kalman filtering. That is, our approximations and biological implementation do not change the computational complexity of these methods. We will clarify this in the revised manuscript.

---

> > ### Comment · Reviewer_pVEa · 2021-09-02
> > **Keep the original score**
> >
> > Thank you for considering this aspect in the revised manuscript. After reading the comments of the other reviewers and the authors' responses, the reviewer will not change his evaluation. The approach is very interesting and the evidence provided supports the authors' claims. Is there room for improvement? Yes, but the work is worthy of publication.

---

### Official Review · Reviewer_yCNb · 2021-07-19

**Rating:** 7
**Confidence:** 4

**Summary:**

The authors propose a model of optimal feedback control for linear dynamical systems that can be realized via biologically plausible learning rules. The authors recognize the importance of integrating both prediction and control and achieve this by integrating two modules -- a model-based state estimation module that combines noisy measurements with internally generated predictions, and a model-free controller module that selects actions based on the estimated state. They demonstrate that the state estimator module can learn the parameters of the system using Hebbian-like weight updates provided the model parameters are initialized appropriately. They also demonstrate that the effect of sensory delay on the proposed model is comparable to what would be expected from an optimal model. The control module uses a standard policy-gradient method to learn a mapping from states to actions. The authors also evaluate their model in a neuroscience-inspired reaching task as well as a new flying task designed in OpenAI gym.

**Limitations And Societal Impact:**

(1) **Lines 256-258**: *Locally linearized dynamics [33] has been suggested to generalize the Kalman filter. The inputs could also be processed using additional neural network layers to obtain a representation that renders the dynamics linear.*  Could you be a bit more specific about how the ideas proposed here would generalize to nonlinear systems while still retaining the biological plausibility?

(2) **Lines 276-278**: *"Our model suggests that system identification is performed by adapting connections between cerebellum and parietal cortex, actions are refined by changes in synapses connecting parietal cortex with premotor cortices".* This is a particularly strong speculation because, to my knowledge, connections from parietal to premotor are not thought to implement model-free computations. The state estimation / controller modules better correspond to cortical / subcortical areas. Which brings me to another question. Perhaps you could also discuss the limitations of using a purely model-free controller?

**Main Review:**

The work is fairly original in its goal of performing system identification using biologically plausible algorithms and integrating it with a model-free controller. The authors have clearly identified how their contribution builds on previous work. The mathematical proofs and techniques are sound but the results fall short of adequately highlighting the consequences of key assumptions and choices proposed in this work. Although biological plausibility is a huge constraint, I find the focus on linear dynamical systems somewhat restrictive particularly since the problems faced by the brain are almost always nonlinear. The authors certainly acknowledge this limitation but they can be more elaborate in pointing the way for readers interested in taking these insights to nonlinear settings. Here are some specific questions and suggestions.

(1) I found the idea of simply replacing $C^\text{T}$ with $L$ really cool. But what are the precise consequences? How would the results shown in Figure 2 look if you used $C^\text{T}$ ?

(2) How does the learning rate and asymptotic performance depend on the degree of initial alignment between $C^\text{T}$ and $L$ ? In general, the authors should better highlight the interaction between algorithmic assumptions and performance.

(3) The authors of ref [8] stated in their study that parameter $C$ could not be learned with Hebbian-like rule. Why does it work here? Was it just a matter of using the right initialization or were there other reasons?

(4) **Supplementary Lines 91-92**: *The marginal mean-squared error decreases if the angle between the gradient gθ for parameter θ and
92 the update ∆θ is less than 90◦, i.e. gθ⊤∆θ > 0.* This certainly explains why the MSE would decrease **initially**. Is there a guarantee that this condition is met **throughout** learning when $C$ and $L$ are learned simultaneously according to the proposed learning rule?

(5) **Figure 3**: Why is model-based state estimation still better than model-free even when delay is 10, given that the trial length is only 10?

(6) **Lines 180-183**: *It shows that a model can be useful in two ways: It facilitates a filtered estimate that is useful even in the absence of measurement delays (see LQG vs model-free for delay 0 in Fig. 3), and in the presence of delays the model helps to bridge the gap by predicting forward in time.* These statements are a tad underwhelming and are true in general, independent of this work. Could you highlight the specific consequences of the approximations made in the proposed model by comparing Bio-OFC against LQG? I could be mistaken, but I did not see any new features in the proposed model that make it robust to sensory delays.

(7) **Lines 208-210**: *Strikingly, our network is capable of closed-loop control with no separate phases for system identification and control optimization necessary*. While it is reassuring to know that the state estimator and the controller could be learned in parallel, why is this unexpected?

(8) **Reaching task (Figure S8)**: Although the task is biologically motivated, it is greatly simplified by assuming that the motor command is already in cartesian coordinates as opposed to controlling the torques applied on joints (which would make the system nonlinear). This is fine, but it means that the correspondence to human behavior is hard to interpret without additional assumptions. Questions: Was there a separate estimator/controller for each reach condition? Was the learning in response to force field driven primarily by changes in the estimator or controller? How does the model learn a time-dependent A?

**Time Spent Reviewing:**

5

---

> ### Author Response · Authors · 2021-08-10
> **Response to reviewer yCNb 1/2**
>
> We are grateful to the reviewer for their detailed review and comments. We will incorporate these suggestions and will include several new plots which we believe will make the paper stronger and more clear.
>
> **1. What are the consequences of replacing $C^\top$ with $L$?**
>
> We reran the experiment in Fig. 2 in the paper (LDS1) using  $C^\top$ as in Eqs. 10 and 11. Interestingly, in this experiment, we found that the learning rate that minimizes the average MSE over all episodes is smaller, convergence is slower, and the average MSE is marginally larger. However, the overshooting after the covariance change at 2500 episodes (in Fig. 2.A) does not occur. We will add these figures to the revised paper.
>
> **2. Dependence of learning rate and performance on initial alignment of $C^\top$ and $L$?**
>
> We investigated this dependence for the LDS1 experiment and find that the performance does not change in a statistically significant way over a wide range of initial alignments that we considered. We will provide plots of the optimal learning rate (found via hyperparameter search) and optimal asymptotic performance as a function of the initial alignment of $C^\top$ and $L$ in the supplementary materials.
>
> In more detail, we learned the Kalman gain $L$ for LDS1 ($C=I$), initializing $L$ as the matrix $\begin{pmatrix} 1-\lambda & \lambda,\\ \lambda & 1-\lambda \end{pmatrix}$, which has eigenvalues $1$ and $1-2\lambda$. If $\lambda=0$ then $L$ and $C^\top$ are perfectly aligned, for $\lambda=0.5$ one eigenvalue of $L$, and thus of the symmetric part of $LC^{\top +}$, becomes zero and performance did not change during learning, for values $\lambda>0.5$ performance deteriorated.
> The asymptotic performance for $\lambda \in [0,.49]$ did not differ in a statistical significant way (p>0.8), but convergence was slower for larger $\lambda\lesssim0.5$.
>
> **3. Why Hebbian learning rules for $C$ work here but fail in [8]?**
>
> We are not completely clear on why [8] fails to learn the $C$ variable. Judging by the paper, it seems the reasons are also not fully clear to the authors. One reason could be that their provided code differs from their derivation presented in the preprint. In the preprint they define $e_x=\mu_{t+1}-A \mu_t-B u_t$, but in the code they substitute $\mu_{t}$ for $\mu_{t+1}$ and use $e_x=\mu_t-A \mu_t-B u_t$ without explanation. For their choice of dt=0.001, the introduced error is small enough to go unnoticed. We verified that for a larger dt, say dt=0.1, even their basic neural Kalman filtering (github.com/BerenMillidge/NeuralKalmanFiltering/blob/master/NKF.ipynb) without any system identification fails, whereas it succeeds if one uses indeed $e_x=\mu_{t+1}-A \mu_t-B u_t$. Further, the different time index $\mu_t$ for updating $A$ (Eq 7) and $\mu_{t+1}$ for $C$ (Eq 9) is also not reflected in their code.
>
> Another reason could be that the authors of [8] simulate a single long sequence, whereas we simulate many trials of a short sequence. The optimization of $C$ minimizes $\|y-C\mu\|^2$. If the estimate $\mu$ is far off from the true $x$, the estimated $C$ that minimizes the least-squares error will also be far off from the true $C$. For a long sequence it's more likely that $\mu$ wanders far away from $x$ (We as well as the authors of [8] initialize $\mu_0$ at the true $x_0$).
>
> Note that while we are not clear on the exact reason why Hebbian learning fails for [8], the fact that it does work in our algorithm is evident in our experiments.
>
> **4. Is there a guarantee that $\mathbf{g}_{\mathbf \theta}^\top \Delta {\mathbf \theta} >0$ throughout learning?**
>
> When performing stochastic gradient descent only the average update aligns with the negative gradient, whereas individual updates could even increase the objective. Similarly $g_\theta^\top \Delta\theta>0$ has to hold only on average. Note that $g_\theta^\top \Delta \theta  = v_{t-\tau} e_t^\top CL e_t v_{t-\tau}^\top$, where $v\in\{\hat{x},u,e\}$ for $\theta\in\{A,B,L\}$. We therefore revisited the simulations of Fig. 4 for delay=1 and kept track of $e_t^\top C L e_t$, which needs to be positive on average. While $e_t^\top C L e_t$ was negative for 8% of the individual updates for LDS2 (0% for LDS1), the average over one epoch $\sum_t^T e_t^\top C L e_t$ was negative for merely 0.04% of the epochs, and always positive if averaged over 10 epochs. Although we do not present a theoretical derivation to show that $\mathbb{E}[e_t^\top C L e_t]>0$ the simulations show this is the case. We will include these simulations in the supplementary materials.
>
> **5. Why is model-based state estimation better than model-free for delay=10 when trial_length=10?**
>
> In our simulations, we either assume that the initial state of the environment is at the true initial state (LDS1,2 and reaching) or that the initial state is random but trial length of the environment with delay=d is preceeded by d steps where the controller is inactive (fly environment). In both cases, the agent has an idea of what state it is in, at the beginning of the control trial. In this way, a trial with length 10 can benefit from model-based state estimation even when delay=10. This is common practice when dealing with delayed environments. We will clarify this in the revised submission.
>
> **6. Statements on lines 180-183 are underwhelming. Highlight specific consequences of the approximations? No new features that make the proposed model robust to sensory delays.**
>
> The reviewer is correct that lines 180-183 are general advantages of performing control via Kalman while considering delay, and not advantages of our model alone. Nonetheless, our contribution is to consistently incorporate these in a biologically plausible algorithm via some approximations that we show are not greatly detrimental to the performance. We will make these points more clear in the revised submission.
> With regards to robustness to sensory delay, we compare proposed Bio-OFC model against LQG in Figure 3. These results are clear demonstration of the fact that the model is robust to sensory delays. Specifically, while it is expected that Bio-OFC will not learn a solution quite as good as LQG, our results show that the solution found by this model not very far off.
>
> **7. Why is closed loop performance of the algorithm unexpected?**
>
> In closed-loop control a controller designs the inputs based on the history of inputs and observations, thus the inputs become highly correlated with the past process noise sequences, which prevents consistent and reliable parameter estimation with standard system identification techniques, see Reference [14].
>
> The work by Linsker [6] is the only other proposed neural implementation that also includes control, but in contrast to our work it requires the separate phases of open-loop control.
>
> By saying "striking" we did not mean that this observation is unprecedented. We are simply pleased that our algorithm can perform closed-loop control and system-ID. We will reword this sentence appropriately.
>
> **8. Reaching task (Figure S8):**
>
> * **Relationship to human behavior is hard to interpret because of the assumptions.** This is true and is a limitation of our work. We will clarify this limitation in the revised draft but leave it for future work to further tighten the connection to biological motor control.
>
> * **Was there a separate estimator/controller for each reach condition?**
>     It is standard in linear control theory to put the target state at the origin. In the reaching task, this can be achieved by using variables related to the difference between the initial state and the target state. This corresponds to remapping the system such that the initial state is on a circle centered at the origin and the target state is at the center.
> As a result, we can use the same estimator/controller for each reach condition and
> the different reach conditions would correspond to different initial states $x_0$.  We will clarify this in the revision.
>
> * **Was the learning in response to force field driven primarily by changes in the estimator or controller?** Learning is driven primarily by changes in the estimator. Even switching off learning in the controller yields similar results. Because the forcefield alters the system, greater changes in the part that performs system identification, i.e. the estimator, are to be expected.
> This is an interesting demonstration and we thank the reviewer for the suggestion. We will provide figures in the supplementary material to showcase this.
>
> * **How does the model learn a time-dependent A?** The model learns the switch from the A of the nullfield to the A of the forcefield via Eq (12). To be clear, in this case the model is not learning time-dependent dynamics. After the switch is made, the model now converges to the solution for system ID of the new dynamics.
>
>     However, if the system changes at a pace much slower than that of the learning dynamics, because of the online nature  of our algorithm, we expect that all system-ID variables ($\hat A,\hat B$, etc.) would track the changes in the system.
>
> Response to comments in limitations section posted separately.

---

> > ### Comment · Reviewer_yCNb · 2021-09-01
> > **increasing score**
> >
> > I'd like to thank the authors for their detailed response. While it is desirable to have a better theoretical understanding of precisely why the approximate learning rule works here and under what circumstances it could fail, I agree that the empirical results reported here are noteworthy by themselves. On a slightly critical note, the final point about the learning dynamics being faster than the system dynamics is likely not true in biology --- synaptic weights probably don't change that fast --- so the interpretation of the force field experiments is still not that convincing. However, I am satisfied with the responses and believe the promised additions would make the paper clearer.
> >
> > I am increasing my score to 7.

---

> ### Author Response · Authors · 2021-08-10
> **Response to reviewer yCNb 2/2**
>
> ## Limitations and societal impact comments
>
>
> * **More specific comments on generalization to non-linear systems while keeping bio-plausibility.** Generalizing our linear learning system to non-linear dynamics is a challenging task and subject of ongoing investigations. However, there are two general approaches that we can take. First, by including an expanded representation in the form of appropriate dictionary functions (e.g. radial basis functions that tile the space), non-linear dynamics can be made linear. A more complex but more efficient approach is to learn the Koopman eigenfunctions of the system. We will include a one paragraph discussion of these approaches in the revised manuscript.
>
> * **Connections from parietal to premotor are not thought to implement model-free computations.**
> Since the reviewer deems this speculation controversial, we have decided to remove this sentence.
>
> * **Discuss the limitations of using a purely model-free controller.** This is a valid point. In the submission we outlined the advantages of having a model-free controller, especially in regards to biological plausibility. We will balance this with a discussion of the draw-backs of model-free controllers, in particular the fact that their efficiency is subpar compared to model-based controllers when the model-class is known.

---

### Official Review · Reviewer_s8QW · 2021-07-19

**Rating:** 7
**Confidence:** 3

**Summary:**

The authors propose a neural network that learns state estimation and control of a linear system via bio-plausible plasticity rules that implement adaptive Kalman filtering and policy gradient. Their scheme incorporates sensory delays, does not need to know noise covariances and can operate in closed loop without separate learning and execution phases.

**Limitations And Societal Impact:**

Yes

**Main Review:**

This paper lucidly introduces neural optimal control, reviews the literature widely, and implements an adaptive neural Kalman filter and controller with bio-plausible learning rules. Their system can handle delays (known a priori) and noise covariances (not known a priori). A key insight in obtaining local learning rules for the Kalman filter is to replace $C^T$ with $L$ albeit requiring prior initialization constraints. The authors learn the controller by a form of REINFORCE - output node perturbation, made bio-plausible using eligibility traces. The authors test their network to control example linear systems, and demonstrate a few advantages of their method -- here, they should clarify why these advantages ensue. While currently limited to linear systems, the authors suggests ways in which this limitation may be overcome. Overall, this implementation simultaneously addresses multiple issues in bio-plausible adaptive neural control, and is clearly exposited. I expect it to enable further cross-fertilization between the control and neural reinforcement learning communities.

**Time Spent Reviewing:**

2

---

> ### Author Response · Authors · 2021-08-10
> **Response to reviewer s8Qw**
>
> We thank the reviewer for their thoughtful review and encouraging comments.
>
> * **Clarify why advantages ensue:**
>     1. The desirable properties of our approach generally follow from the optimal combination of noisy observations and the predictions of the internal model which follows from the implementation of the Kalman filter. In the case of delay, these advantages are exaggerated because of the compound effects of noise in the observed state.
>     2. The advantages of our method vs. prior work (tabulated in Tab. 1), were generally described in Sec. 1 and Sec. 2 of the manuscript.
>
>     We will expand both of these points further in the final version of the submission such that the origin of each improvement claim is clear. If the reviewer finds that there are specific points that would benefit from further explanation, we would be happy to expand on it in the revised manuscript.

---

### Author Response · Authors · 2021-08-10
**General response**

We thank the reviewers for their thoughtful feedback. We are encouraged that they found our submission clear [*s8QW, pVEa*]
,
novel and original [*yCNb, pVEa*]
, and
enabling cross-fertilization between control and neural reinforcement learning communities [*s8QW*].
We are also pleased that the reviewers
appreciate the derivation of the bio-plausible learning rules by replacing $C^\top$ with $L$ [*s8QW,yCNb*].

We appreciate the overall positive reaction of the reviewers and the constructive feedback, which we will address individually.

---

### Decision · Program_Chairs · 2021-09-27

**Decision:**

Accept (Spotlight)

**Comment:**

This paper had uniform support from the reviewers, and is a clear accept. To improve the work, I highly suggest you:
1 . Add the additional experiment requested by a reviewer
2. More clearly outline the derivation for the update equations (we discussed this in our discussion and concluded that they were correct, but it was not immediately obvious just from the text)
3. Incorporate other suggestions by the reviewers.